# Discovery of fungal surface NADases predominantly present in pathogenic species

Øyvind Strømland[1], Juha P. Kallio [1], Annica Pschibul[2], Renate H. Skoge [3], Hulda M. Harðardóttir[3], Lars J. Sverkeli[1,3], Thorsten Heinekamp[2], Olaf Kniemeyer[2], Marie Migaud[4], Mikhail V. Makarov[4], Toni I. Gossmann[5,6], Axel A. Brakhage[2,7] & Mathias Ziegler [1✉]

Nicotinamide adenine dinucleotide (NAD) is a key molecule in cellular bioenergetics and signalling. Various bacterial pathogens release NADase enzymes into the host cell that deplete the host's NAD$^+$ pool, thereby causing rapid cell death. Here, we report the identification of NADases on the surface of fungi such as the pathogen *Aspergillus fumigatus* and the saprophyte *Neurospora crassa*. The enzymes harbour a tuberculosis necrotizing toxin (TNT) domain and are predominately present in pathogenic species. The 1.6 Å X-ray structure of the homodimeric *A. fumigatus* protein reveals unique properties including N-linked glycosylation and a Ca$^{2+}$-binding site whose occupancy regulates activity. The structure in complex with a substrate analogue suggests a catalytic mechanism that is distinct from those of known NADases, ADP-ribosyl cyclases and transferases. We propose that fungal NADases may convey advantages during interaction with the host or competing microorganisms.

[1] Department of Biomedicine, University of Bergen, Bergen, Norway. [2] Department of Molecular and Applied Microbiology, Leibniz Institute for Natural Product Research and Infection Biology, Hans Knöll Institute, Jena, Germany. [3] Department of Biological Sciences, University of Bergen, Bergen, Norway. [4] Mitchell Cancer Institute, University of South Alabama, Mobile, AL, USA. [5] Department of Animal Behaviour, Bielefeld University, Bielefeld, Germany. [6] Department of Animal and Plant Sciences, University of Sheffield, Sheffield, UK. [7] Institute of Microbiology, Friedrich Schiller University Jena, Jena, Germany. ✉email: mathias.ziegler@uib.no

Since its discovery in the 1930s[1], nicotinamide adenine dinucleotide (NAD$^+$) has emerged as one of the fundamental metabolic molecules of life and as a central player in cellular signalling processes owing to its function as substrate for various ADP-ribose transfer reactions[2,3]. Not long after NAD$^+$ was discovered, enzymatic hydrolysis of the dinucleotide was detected. The enzymes that break the N-glycosidic bond between the terminal ribose and the nicotinamide (Nam) moiety are termed NAD$^+$ glycohydrolases or NADases[4–6]. In addition to cleavage of NAD$^+$ to Nam and ADP-ribose, NADases found in animals act as ADP-ribosylcyclases, generating cyclic ADP-ribose (cADPR), a potent intracellular Ca$^{2+}$ mobilizing agent[7]. However, in microbial species, NADases appear to have evolved as powerful weapons to mediate infections[5,6,8]. For example, *Streptococcus pyogenes* and *Mycobacterium tuberculosis* produce enzyme toxins that trigger necrotic cell death of host macrophages by rapid NAD$^+$ depletion. Recently, it was discovered that certain Toll/interleukin-1 receptor (TIR) domains in animals and plants constitute a new family of NADases, which are involved in cell death pathways and defence against infections[9–11]. For example, mammalian SARM1 is an NADase that mediates axon degeneration through rapid NAD$^+$ degradation[9].

Even though ubiquitously found in bacteria, plants and animals no NADase has so far been identified in fungi. Earlier reports indicated the presence of NADase activity on conidia and hyphae of the filamentous fungus *Neurospora crassa*[12,13]. However, the molecular identity of this activity has remained obscure. Given the widespread occurrence of NADases and their prominent role in microbial infection mechanisms we wondered whether pathogenic fungi may also produce similar enzymes.

In this work, we identify NADases on the surface of fungi and show that these enzymes are predominantly found in pathogenic species. The *A. fumigatus* and *N. crassa* enzymes cleave NAD$^+$ and NADP$^+$ but not their reduced counterparts, NADH and NADPH. Moreover, the enzymes lack both ADP-ribosyl cyclase and base exchange activity. The structure of the dimeric *A. fumigatus* enzyme revealed the presence of a Ca$^{2+}$ binding site whose occupancy partially regulates enzymatic activity. In addition, each protomer is N-linked glycosylated at three asparagine residues and stabilized by two disulphide bridges. We also solved the structure of the enzyme bound to the reaction products, nicotinamide and ADP ribose, and the non-hydrolysable substrate analogue benzamide adenine dinucleotide (BAD). The structure bound to the substrate analogue along with mutagenesis of predicted critical amino acid residues led us to suggest a reaction mechanism distinct from all known NADases. We propose that these NADases may represent hitherto unrecognized factors that convey advantages for the fungi during interaction with the host or competing microorganisms in the environment.

## Results and discussion

**Identification of fungal surface NADases.** Since conidia from *N. crassa* have been known to exhibit NADase activity, we tested whether conidia from the human opportunistic pathogen *A. fumigatus* may have a similar activity[13]. Using the fluorescent NAD$^+$ analogue nicotinamide 1, N6-ethenoadenine dinucleotide (εNAD$^+$) we detected robust cleavage on the surface of *A. fumigatus* conidia from the strain CEA17ΔakuB, indicating the presence of one or more NADases (Fig. 1A). NADase activity was present at different growth stages suggesting that one or several surface NADase(s) are expressed during *A. fumigatus* development (Fig. 1B). NADase activity was also present on conidia from the clinical strains Af293, D141 and ATCC46645 (Supplementary Fig. 1A). Since the surface proteome of *A. fumigatus* is dependent on the nutritional source[14], we investigated whether the activity

was influenced by the growth medium. NAD$^+$ cleavage activity was present on conidia cultivated in all media tested, with higher activity found using *Aspergillus* minimal media and malt agar compared to Saboraud and RPMI agar (Supplementary Fig. 1B). $^1$H NMR showed that the conidial enzyme(s) cleave(s) NAD$^+$, yielding ADP ribose (ADPR) and Nam as reaction products (Fig. 1C). Thereby, we established the presence of one or more NADases on the surface of conidia from *A. fumigatus*. To identify the conidial protein(s) of *N. crassa* and *A. fumigatus* responsible for NAD$^+$ cleavage, we first visualized their activity in a polyacrylamide gel following SDS-PAGE in non-reducing conditions using εNAD$^+$ [15]. Conidia from both *N. crassa* and *A. fumigatus* exhibited strong NADase activity with a migration corresponding to a molecular mass of ~50 kDa (Fig. 1D). Strikingly, NADase activity was still detectable in the *A. fumigatus* conidia following heat treatment at 95 °C for 5 min, albeit with the majority of the detectible activity migrating as a protein with a molecular mass of ~30 kDa (Fig. 1D). LC-MS/MS-based proteomic analyses of the bands exhibiting NADase activity identified 27 proteins in untreated *A. fumigatus* conidia compared to eight in heat-treated conidia (Supplementary Data 1). The only overlapping hit between the two samples was the predicted gene product AFUA_6G14470, a protein with unknown function and a theoretical molecular mass of 26 kDa. Of note, this protein has already been detected previously on the surface of conidia[14,16]. Analyses of conidia from *N. crassa* yielded peptides corresponding to a hypothetical protein homologous to that identified in *A. fumigatus* (Fig. 1E). Bioinformatic analyses of the sequences predicted three N-linked glycosylated asparagine residues with high confidence (Fig. 1E). This is in line with studies that showed glycosylation of *N. crassa* NADase and could explain the discrepancy between the observed and theoretical molecular masses[13]. A predicted N-terminal secretory signal peptide in the *A. fumigatus* protein was confirmed by LC-MS/MS (highlighted in blue in Fig. 1E). To verify the identity of the suspected NADase gene, we generated an *A. fumigatus* knock-out strain lacking the gene AFUA_6G14470. Indeed, we did not observe any NADase cleavage in conidia of this knock-out strain, whereas in-locus complementation restored NADase activity (Fig. 1F, G). These results confirmed that the gene AFUA_6G14470, now named *nadA*, encodes a conidial NADase in *A. fumigatus* (GenBank accession number: MT276230). Moreover, we have established the molecular identity of the long-sought *N. crassa* NADase, encoded by the gene NCU07948 (GenBank accession number: MT316195).

**Fungal NADases hydrolyse NAD$^+$ and NADP$^+$, but do not mediate synthesis of calcium messengers (cADPR or NAADP).** Next, we investigated the catalytic and proteo-chemical properties of the fungal NADases. Recombinant *A. fumigatus* NADase (*Af*NADase) was expressed in stably transfected human 293 cells as well as recombinant baculovirus-infected Sf9 insect cells. In both systems, the overexpressed enzyme was secreted into the medium and exhibited robust NADase activity (Fig. 2A and Supplementary Fig. 2A, B). The purified proteins, expressed in Sf9 and 293 cells, exhibited a size of ~30 kDa and 40 kDa, respectively, based on their migration in SDS-PAGE, presumably owing to different extent of glycosylation. Size-exclusion chromatography suggested the enzyme to be homodimeric (Supplementary Fig. 2C–F). Subsequent experiments were performed with the protein expressed in and purified from Sf9 cells. As observed with conidia, *Af*NADase activity was unusually heat-resistant (Tm = 78.5 °C) and could be detected following both 5 and 10 min of incubation at 95 °C (Supplementary Fig. 2G, H). However, the enzyme was sensitive to the reducing agent DTT and the metal chelator EGTA. Inhibition

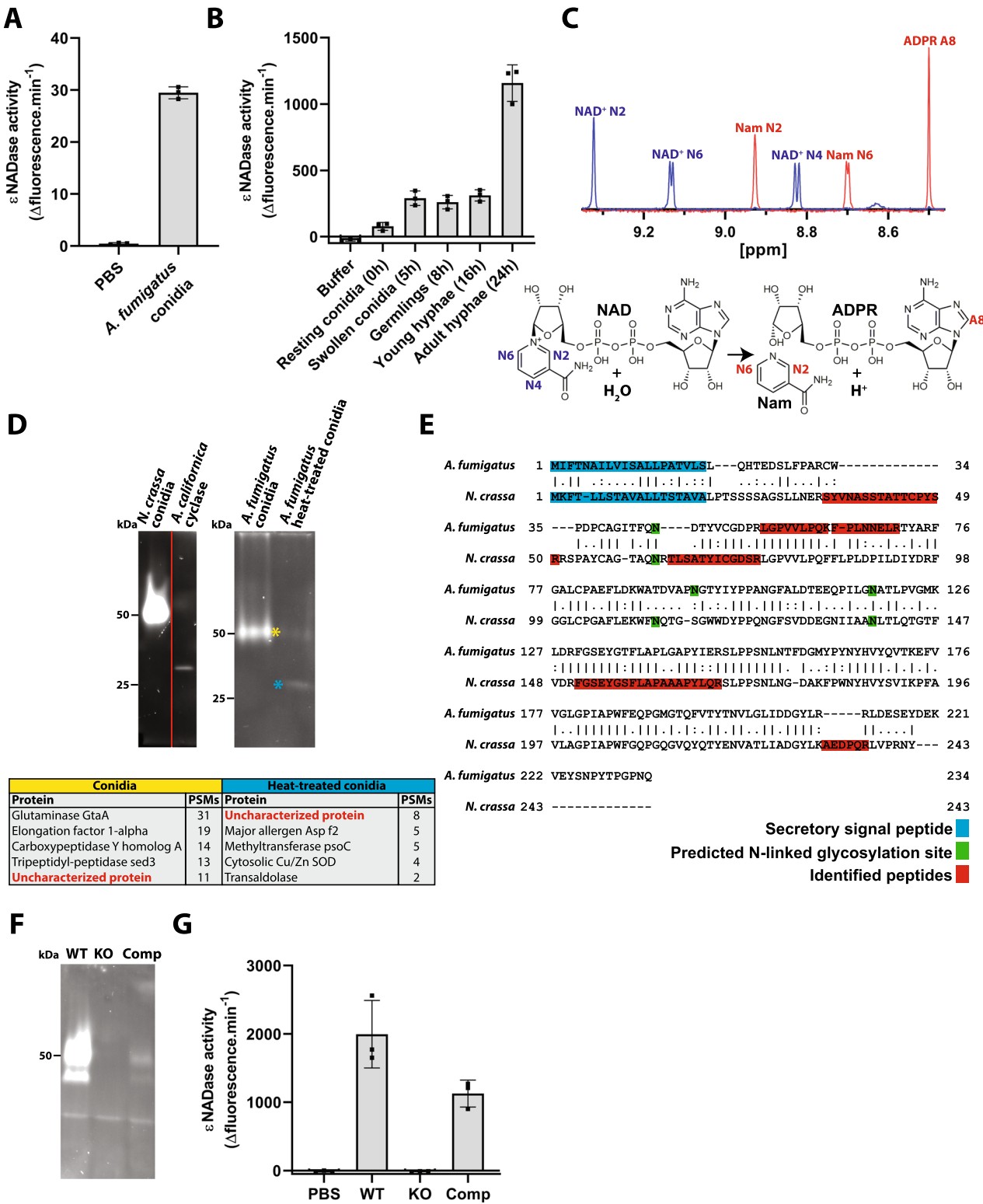

by EGTA was reversed by $Ca^{2+}$ (Fig. 2B), but not by other bivalent metal ions such as $Mn^{2+}$, $Mg^{2+}$ or $Zn^{2+}$. EGTA titration showed that the enzyme is only partially dependent on $Ca^{2+}$, as activity was still detected even in the presence of 5 mM EGTA. Interestingly, *N. crassa* NADase (*Nc*NADase) activity was not affected by either EGTA or $Ca^{2+}$ (Supplementary Fig. 2I). Both *Af*NADase and *Nc*NADase cleaved $NAD^+$ and $NADP^+$, but not their reduced counterparts, NADH and

NADPH (Fig. 2C, Supplementary Figs. 3, 4). Besides hydrolysis of $NAD(P)^+$ to (phospho)ADPR and Nam, animal NADases (ADP-ribosylcyclases) also catalyse the formation of the $Ca^{2+}$ messenger cyclic ADPR (cADPR) as well as the exchange of the nicotinamide moiety by nicotinic acid (Fig. 2E). Both *Af*NADase and *Nc*NADase did not produce any detectable cADPR and did not mediate base-exchange with nicotinic acid (NA), reactions readily catalysed by *Aplysia californica* cyclase

**Fig. 1 Identification of fungal conidial NADases. A** NADase activity of *A. fumigatus* conidia demonstrated by a fluorometric assay using εNAD, $n = 3$.
**B** NADase activity of different *A. fumigatus* growth stages demonstrated by a flurorometric assay using εNAD, $n = 3$. **C** Top: identification of nicotinamide (Nam) and ADP ribose (ADPR) as the $NAD^+$ cleavage products following incubation with *A. fumigatus* conidia by $^1$H NMR. Bottom: Assignment of relevant NMR signals for $NAD^+$ and its cleavage products. Hydrolysis of $NAD^+$ leads to the formation of Nam and ADPR, the protons giving rise to the NMR signals are labelled blue for $NAD^+$ and red for Nam and ADPR. **D** Identification of possible fungal NADases. Top: Enzyme activity gels of *N. crassa* and *Aplysia californica* cyclase, which was used as a positive control (left) and *A. fumigatus* (right) conidia developed with εNAD. In the right panel, the yellow and cyan asterisks denote the bands obtained with and without heat treatment, respectively. Fluorescent bands were excised and subjected to proteomics analysis. Bottom: Top hits of *A. fumigatus* proteins identified by mass spectrometry sorted by their peptide spectral matches score (PSM). The only overlapping hit between the two samples was an uncharacterized, predicted protein (highlighted in red). **E** Global pairwise alignment of predicted NADase sequences from *A. fumigatus* and *N. crassa* deduced from the genes identified based on the results shown in (**D**). The peptides that were detected in *A. fumigatus* and *N. crassa* conidia are highlighted in red. Bioinformatic analyses predict the presence of a secretory signal peptide and N-linked glycosylated asparagine residues, shown in blue and green, respectively. **F** In-gel εNADase assay of conidia from *A. fumigatus* wild type (WT), knockout mutant (KO) strain *ΔnadA* which lacks the gene predicted to encode the conidial NADase (AFUA_6G14470) and conidia from a complementation strain (Comp) *ΔnadA:nadA*.
**G** Fluorometric assay of NADase activity of the samples described in (**F**), $n = 3$. Experiments in (**A, B, D, F, G**) were performed independently three times with similar results. Source data are provided as a Source Data file.

(Fig. 2F). *Af*NADase also failed to hydrolyse nicotinic acid adenine dinucleotide (NAAD), the deamidated form of $NAD^+$ (Supplementary Fig. 5A). The absence of ADP-ribosyl cyclase activity from the fungal NADase was further substantiated by its inability to convert the $NAD^+$ analogue nicotinamide hypoxanthine dinucleotide ($NHD^+$) to Nam and fluorescent N7-cyclic inosine diphosphoribose (N7-cIDPR), a reaction specifically observed for cyclases (Supplementary Fig. 5B–D). These results demonstrate that fungal NADases are pure NAD glycohydrolases as they lack both cyclase and base-exchange activity. We determined the kinetics of *Af*NADase-mediated $NAD^+$ and $NADP^+$ hydrolysis by $^1$H NMR. The hydrolysis of $NAD(P)^+$ was followed by inspecting the resonance decay of $NAD(P)^+$ protons and the corresponding resonance increase in Nam and (2'-phospo-) ADPR protons (Fig. 2D). The $K_M$ were calculated to be $119.7 \pm 40.8\,\mu M$ and $106 \pm 27.1\,\mu M$ for $NAD^+$ and $NADP^+$, respectively. The turnover rates were determined to be $1962 \pm 133\,s^{-1}$ and $418 \pm 108\,s^{-1}$ for $NAD^+$ and $NADP^+$, respectively.

**The crystal structure of AfNADase reveals the presence of a TNT domain and distinct structural properties including a regulatory calcium-binding site.** Next, we sought to get insights into the structural assembly and the catalytic mechanism of *Af*NADase. We solved the crystal structure of *Af*NADase to a resolution of 1.6 Å (Fig. 3A and Supplementary Table 1) (PDB: 6YGE). The protein crystallizes in the space group P3221 with two molecules in the asymmetric unit confirming the homodimeric assembly suggested from the chromatographic analysis (Supplementary Fig. 2C, D). The dimeric assembly is formed via an interface of 2260 Å$^2$ of the total solvent-accessible area of 11030 Å$^2$ (calculated with PDBePISA server), involving 66 residues, including the C-terminus which is intertwined with the other protomer (Fig. 3A). The protomers consist of two domains, an N-terminal 'thumb' and a C-terminal 'palm' domain. The thumb domain, residues 20-117, is folded containing five α-helices connected with loop regions, and the fold is stabilized by two disulfide bridges, C33-80 and C38-50. The palm domain consists of a seven-stranded central β-sheet flanked by two short α-helices and two 3$_{10}$ helices. As described below, we identified the palm domain as TNT domain, based on the 3D structure. In addition, the C-terminus contains a metal binding site. The location of this binding site is at the dimerization interface, just before the C-terminus turns to intertwine with the other protomer. Noteworthy, an acetate ion, originating from the crystallization solution, is trapped in a cavity located near the domain boarder and facing the central β-sheet (strands β2, 3 and 5).

The pentagonal bipyramidal coordination of the metal ion in the crystal structure, the refined density as well as the B-factors indicate that the bound metal ion is $Ca^{2+}$, in line with the observed $Ca^{2+}$ activation of the enzyme following EGTA treatment (Fig. 2B). The $Ca^{2+}$ ion binding site is comprised of the side chains of D219, E220 and E223 as well as the main chain of S216. In addition, two water molecules are involved in metal ion coordination (Fig. 3A, C). Each protomer is glycosylated at three asparagine residues, N45, N95 and N118 (Fig. 3B) and these sites coincide with those originally predicted (Fig. 1E). N-linked glycosylation of *Nc*NADase expressed in 293 cells was also observed (Supplementary Fig. 6). Taken together, these observations provide the structural basis for the biochemical findings regarding DTT and EGTA sensitivity, as well as migration in SDS-PAGE according to a molecular mass higher than predicted from the polypeptide alone (Fig. 1D). To elucidate a possible role of the $Ca^{2+}$ binding site of *Af*NADase in catalysis, we created a $Ca^{2+}$-free mutant by substituting D219 and E220 with alanine residues. Disruption of the $Ca^{2+}$ binding site by mutagenesis led to a sevenfold reduction in $εNAD^+$ hydrolysis as compared to the WT enzyme, highlighting the importance of $Ca^{2+}$ coordination for efficient hydrolysis (Fig. 3C, D). To get a better understanding of the $Ca^{2+}$ ion in catalysis we attempted to crystallize the $Ca^{2+}$ binding site mutant of *Af*NADase, however the protein did not yield diffraction quality crystals. To circumvent this issue, we crystallized the native protein treated with EGTA. However, $Ca^{2+}$ was still present in the crystal structure, albeit with a lower occupancy. Interestingly, when these crystals were soaked with a tenfold molar excess of $NAD^+$, the $Ca^{2+}$ ions were no longer present in the crystal, and in molecule A of the asymmetric unit the reaction products, Nam and ADPR, were bound to a putative active site (Fig. 4A, B) (PDB: 6YGF). Nam is located in the deep cavity, where acetate was trapped in the apo structure, and is hydrogen-bonded to the enzyme with the side chains of R129 and R148, the main chain of F130 and a water molecule, from here on referred to as water I (Fig. 4C). These residues are highly conserved, suggesting they play a role in hydrolysis of $NAD(P)^+$ (Supplementary Fig. 7). ADPR is found in close proximity to Nam and rests in a crevice on the surface of the enzyme. The ribose moiety that was bound to Nam is flipped out of the cavity and makes hydrophobic contact with the aromatic ring of Y100, a residue that is also conserved. The two phosphate moieties make extensive contacts with the enzyme through hydrogen bonds with R129, S132, L138, N154 and water I. Finally, the adenosine moiety is bound to the enzyme via π-stacking with the side chain of F158. These observations indicate that the residues interacting with Nam and ADPR indeed constitute the active site of *Af*NADase. However, given that the distance between the $Ca^{2+}$

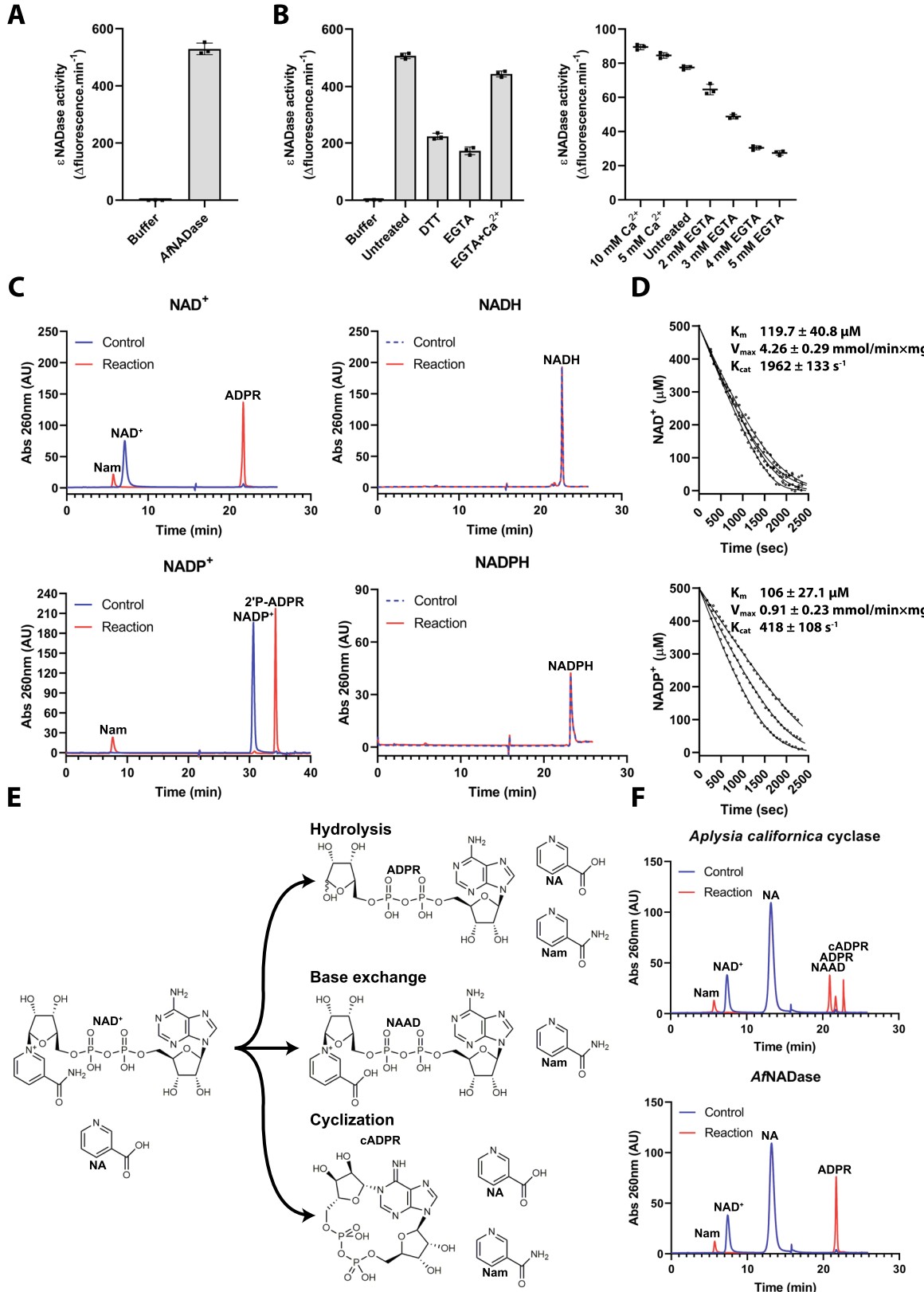

ion and the active site is more than 40 Å, the role of $Ca^{2+}$ in catalysis cannot be easily explained. Since the activity is lower in the $Ca^{2+}$ binding site mutant, it is tempting to speculate that the $Ca^{2+}$ ion plays a role in regulation of the enzymatic activity to prevent suicidal $NAD(P)^+$ depletion as the intracellular concentration of $Ca^{2+}$ is in the nanomolar range in conidia[17].

**Identification of the catalytic mechanism of fungal and other TNT-containing NADases based upon the crystal structure in complex with a non-cleavable substrate analogue.** To understand the catalytic mechanism of AfNADase and related NADases such as TNT, we attempted to solve the structure of the enzyme in complex with $NAD^+$ by soaking the crystals with the

**Fig. 2 AfNADase is a pure NAD(P)$^+$ glycohydrolase. A** NADase activity of recombinant AfNADase purified from Sf9 insect cells measured by the fluorescence assay using εNAD, $n = 3$. **B** left: NADase activity of recombinant AfNADase from Sf9 insect cells treated with DTT, EGTA or EGTA and calcium chloride, $n = 3$ Right: NADase activity of AfNADase from Sf9 insect cells titrated with EGTA and CaCl$_2$. **C** identification of AfNADase substrate specificity by HPLC. The HPLC chromatograms display AfNADase mediated reactions using the indicated substrates. **D** Kinetics of NAD$^+$ and NADP$^+$ hydrolysis by AfNADase determined by $^1$H NMR. The integral of the resonances corresponding to NAD$^+$ N-2 and N-6 were plotted and used for curve fitting. **E** Absence of ADP-ribosyl cyclase activity in AfNADase. In the presence of an excess of NA some NADases (namely ADP-ribosyl cyclases) can catalyse the exchange of the Nam moiety in NAD$^+$ for NA producing NAAD ("base-exchange reaction"). Cyclases can also produce cyclic ADPR or simply cleave NAD$^+$ to ADPR and Nam. **F** AfNADase does not possess base-exchange or ADPR cyclase activity, identifying it as pure NADase. HPLC chromatograms of AfNADase and Aplysia californica cyclase base exchange reactions. Experiments in (**A**, **B**, **C**, **F**) were performed independently three times with similar results. Source data are provided as a Source Data file.

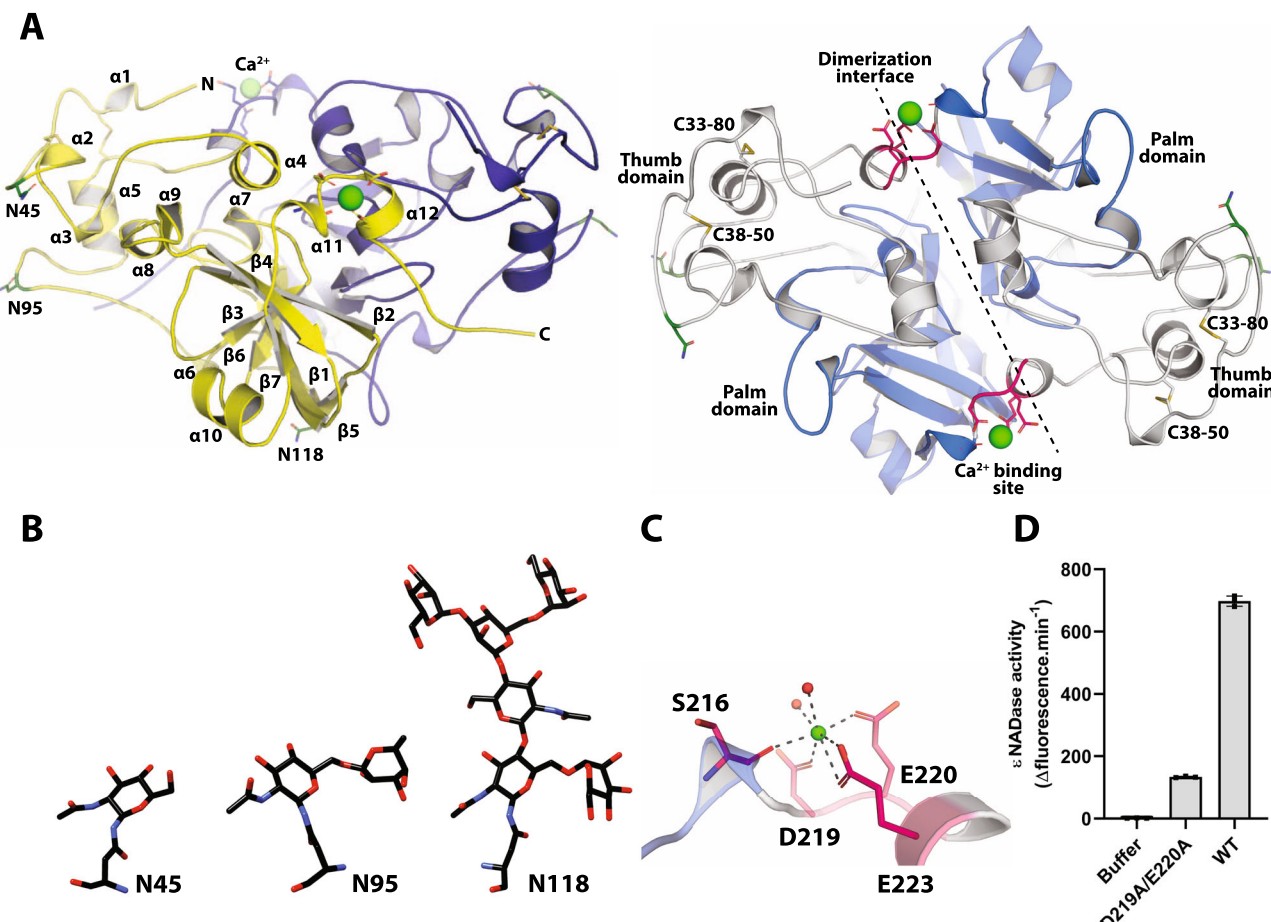

**Fig. 3 Crystal structure of AfNADase. A** Left panel, cartoon representation of AfNADase. The protomers forming the dimer are coloured yellow and blue. The secondary structure elements are sequentially labelled, α and β for helical elements and for beta strands, respectively. The N-linked glycosylated asparagine residues are shown in stick representation. The green sphere represents the bound calcium ion and the residues that are involved in the coordination are shown in stick representation (pink). Right panel, cartoon representation of domain structure (rotated 70° on X-axis from the left panel orientation). The thumb domain is shown in grey and palm domain in blue, and the dimerization interface through the 2-fold symmetry axis is displayed as a dotted line. The disulfide bridges are shown in stick representation. **B** Stick representation of the observable N-linked glycosylation at asparagine 45, 95 and 118. **C** Cartoon and stick representations of the calcium binding site showing the residues and water molecules that are involved in the pentagonal bipyramidal coordination of the metal ion. **D** NADase activity of A. fumigatus calcium binding site mutant (D119A/E220A) measured by the fluorometric assay using εNAD, $n = 3$. The experiment was performed independently three times with similar results. Source data are provided as a Source Data file.

dinucleotide. However, due to the fast conversion rate of NAD$^+$ to Nam and ADPR, it was not possible to trap the substrate in the crystals. We therefore used the non-cleavable NAD$^+$ analogue BAD, differing from NAD$^+$ by the substitution of the nitrogen atom in the pyridine ring with a carbon atom (Fig. 4D). We were able to co-crystallize AfNADase with BAD bound to the active site of molecule B of the asymmetric unit (Fig. 4E, F) (PDB: 6YGG). Residual electron density can be observed in molecule A but not enough to model BAD into the active site. The structure

of the complex clearly illustrates the binding mode of the substrate to the active site, and many of the residues are the same as those involved in product binding (Fig. 4G). The interaction with BAD includes two water molecules; one is found in the same position as water I in the structure of AfNADase bound to the products and the other, water II, makes contact with the proximal ribose (Fig. 4G). In the complex, F137 makes contact with the proximal ribose of BAD positioning the scissile bond into the active site. The benzamide (nicotinamide, by analogy) is also held

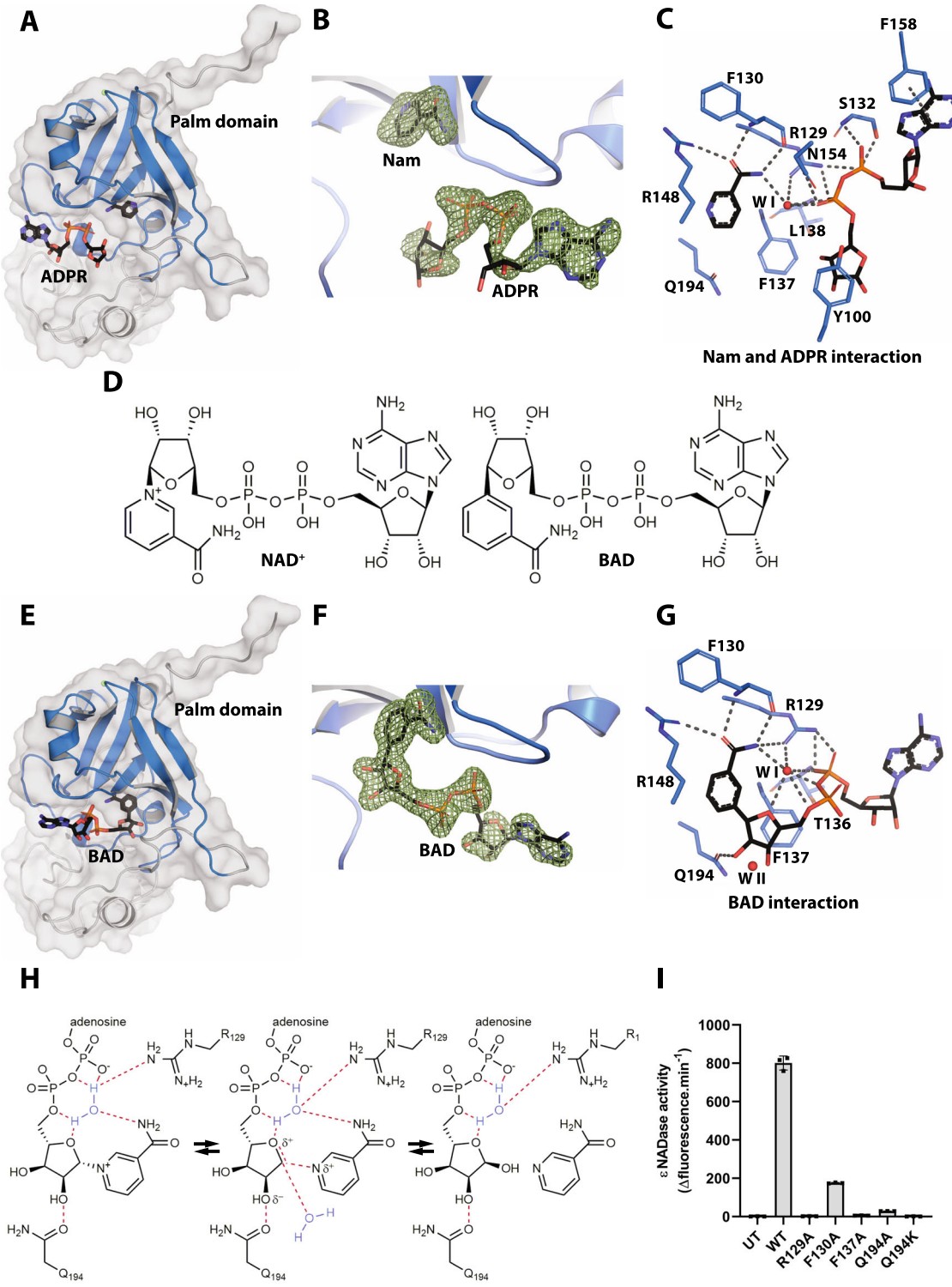

in place by hydrogen bonding to R129, F130 and R148. The sidechain of Q194 makes contact with the 2" OH group of the proximal ribose. In other NADases, ADPR transferases (ARTs), which catalyse the transfer of an ADPR moiety from $NAD^+$ to an acceptor, as well as in ADP-ribosyl cyclases, a catalytic acidic residue, glutamate or aspartate, is found in the corresponding position. In these enzymes $NAD(P)^+$ is cleaved through the formation of an oxocarbenium intermediate. However, this is unlikely to be the case in AfNADase as the pull from Q194 on the bond would not be strong enough. Since the formation of an oxocarbenium reaction intermediate is required for both cyclase and base exchange activity, this observation explains why AfNADase cannot catalyse these reactions. In order to achieve cleavage of $NAD(P)^+$, a nucleophilic attack on the anomeric carbon (C1') is needed. In AfNADase, the attack cannot come from the β-face because this is the location of the nicotinamide moiety and the β-face is shielded by its hydrophobic interaction with F137. We propose a mechanism where water molecule I is hydrogen-bonded to the in-ring oxygen of the proximal ribose, the nicotinamide moiety, the phosphodiester backbone and R129. This could act as a bridge to stabilize a positive charge on the in-ring oxygen. The nucleophilic attack is further prepared by Q129

**Fig. 4 Crystal structure of *Af*NADase in complex with reaction products or the non-hydrolysable NAD+ analogue BAD. A** Complex of *Af*NADase and the reaction products Nam and ADPR. The protein is shown as a cartoon embedded into a transparent surface map. The palm domain is coloured blue and the reaction products Nam and ADPR are displayed in stick representation. **B** Calculated mFo-DFc POLDER electron density map contoured at 3.0 σ to confirm the binding of ADPR and nicotinamide. **C** Residues of AfNADase involved in the interaction with the reaction products Nam and ADPR are shown in stick representation. The water molecule participating in the interaction is labelled WI. The dotted lines represent hydrogen bonds between the protein, water molecule and the reaction products. **D** In comparison with NAD+ the nitrogen in the pyridine rings has been substituted with a carbon atom in the non-hydrolysable analogue BAD. **E** Complex of *Af*NADase and the substrate analogue BAD. The protein is shown as a cartoon embedded into a transparent surface map. The palm domain is coloured blue and BAD is shown in stick representation. **F** Calculated mFo-DFc POLDER electron density map contoured at 3.0 σ to confirm the binding of BAD. **G** Residues of *Af*NADase involved in the interaction with BAD are shown in stick representation. The water molecules participating in the interaction are labelled WI and WII. The dotted lines represent hydrogen bonds between the protein, water molecules and the substrate analogue. **H** Proposed reaction mechanism of *Af*NADase-mediated NAD+ hydrolysis. In an SN2-like reaction mechanism, water molecule I is hydrogen-bonded to the in-ring oxygen of the proximal ribose, the nicotinamide moiety, the phosphodiester backbone and R129. This acts as a bridge to stabilize a positive change on the in-ring oxygen. The nucleophilic attack is further prepared by Q194, which interacts with the 2″ OH of the proximal ribose and induces a δ-negative charge. The attack on the C1′ position from water II coincides with the nicotinamide leaving the active site and the formation of an oxonium intermediate. **I** activity of recombinant *Af*NADase WT and active site mutants R129A, F130A, F137A, Q194 and Q194K measured by the fluorometric assay using εNAD, n = 3. The experiment was performed independently three times with similar results. Source data are provided as a Source Data file.

which interacts with the 2″ OH of the proximal ribose and can potentially induce a δ-negative charge. This mechanism would resemble a SN2 reaction with the formation of a transition state, in which the attack on the α-face on the C1′ position from water II coincides with the nicotinamide leaving and the formation of a partial oxocarbenium ion (oxonium intermediate) (Fig. 4H). The proposed mechanism precludes NAAD cleavage since the interaction between the carboxylate of the nicotinic acid moiety, watermolcule I and the side chain of R129 is less likely to lead to catalysis, in line with our observation that *Af*NADase does not cleave NAAD (Supplementary Fig. 5A). In support of the proposed catalytic mechanism, mutagenesis of R129, F137 and Q194 inactivated the enzyme (Fig. 4I). Mutagenesis of F130 did not completely inactivate the enzyme. This is not surprising given that F130 interacts with the product/substrate with the carbonyl and amide backbone atoms (Fig. 4H). Substitution of R148, which interacts with the nicotinamide (moiety), was also attempted. However, neither Western blotting nor activity measurements indicated the presence of the recombinant protein expressed in 293 cells, suggesting that the protein is unstable and rapidly degraded. In line with this supposition, R148 forms a salt bridge with D128 that may be vital for protein stability. Indeed, mutation of the corresponding R780 in TNT also resulted in an unstable protein (18). These residues are conserved in all TNT domain-containing proteins in both fungi and bacteria (Supplementary Fig. 7).

**Fungal NADases are predominantly present in pathogenic species.** Next, we investigated the phylogenetic relationship between *Af*NADase and other NADases. Initial BLAST searches using the primary structure of *Af*NADase and standard parameters did not return any known NADases. However, a Pfam search revealed that the palm domain of *Af*NADase, residues 117-234, harbours a tuberculosis necrotizing toxin (TNT) domain (Fig. 5A). Intriguingly, the TNT domain was recently identified in the *M. tuberculosis* (*Mtb*) protein CpnT as a toxin with NADase activity, which mediates toxicity towards host macrophages by depleting NAD(P)+. The structure of Mtb CpnT TNT, from here on referred to as TNT, has been solved in complex with its immunity factor, an endogenous inhibitory protein which prevents suicidal NADase activity (PDB: 4QLP)[8]. No homologue of the gene encoding the *M. tuberculosis* immunity factor was found in the genomes of *N. crassa* and *A. fumigatus*. An immunity factor may not be needed in fungi since the enzyme enters the secretory pathway and secretion precludes it from interacting with the major cellular NAD+ pools. Another possible explanation is that *Af*NADase activity is regulated by Ca²⁺ and therefore

an immunity factor is not required. However, we noted certain structural similarities between *Af*NADase and TNT (Supplementary Fig. 8). The palm domain is similar between *Af*NADase and TNT and they can be superimposed with an RMSD of 0.83 Å (Supplementary Fig. 8). However, the thumb domain of *Af*NADase differs markedly from the thumb domain of TNT. In *Af*NADase, this region is stabilized by two disulfide bridges and contains two N-linked glycosylation sites, N45 and N95; features that are not present in TNT. The most distinct feature of the *Af*NADase palm domain is the C-terminal extension harbouring the Ca²⁺ binding site. In *Af*NADase the C-terminus is intertwined with the other protomer, whereas the C-terminus of TNT is located immediately after the last helix in the core fold (helix 10 in *Af*NADase). The active site of TNT coincides with that of *Af*NADase and critical residues and sequence motifs are conserved (Supplementary Figs. 7 and 8). *Af*NADase and TNT exhibit the same substrate specificity and both lack cyclase and base exchange activity (Fig. 2 and Supplementary Fig. 5)[18]. Therefore, the fungal NADases can be classified as TNT domain-containing proteins, which are widely found in bacteria and fungi. Moreover, the proposed reaction mechanism of *Af*NADase is likely valid for all TNT domains.

The fungal NADases also share commonalities with more distantly related NADases and ADP-ribosyltransferases (ARTs), such as Tse6, cholera toxin and diphtheria toxin, as their active sites can be aligned (Fig. 5B, C and Supplementary Table 2). ARTs are divided into two groups based on active site motifs distributed across three regions. The cholera toxin has an R-S-E motif, whereas diphtheria toxin has an H-Y/Y-E motif, which is also present in poly (ADP-ribose) polymerases (PARPs). NADases contain similar sequence motifs and in *Af*NADase the motif consists of R129, F137 and Q194 residues that are vital for NADase activity.

Bacteria and fungi with a predicted TNT domain protein mainly belong to the phyla Firmicutes, Actinobacteria and Ascomycota. However, the fungal NADases exhibit specific structural and functional properties, distinct from their bacterial counterparts. These properties include the expression on the outer surface of the conidia and hyphae, the homodimeric nature, a structurally unique thumb domain, posttranslational glycosylation as well as the presence of critical disulfide bonds. In addition, in some species such as *A. fumigatus*, the catalytic activity is regulated by Ca²⁺ ions.

The Ca²⁺ binding site found in *Af*NADase emerged in the order Eurotiales, which includes *Aspergillus spp* (Fig. 5D). The predicted 3D structure of *Nc*NADase is highly similar to that of *Af*NADase (Supplementary Fig. 9), except that it lacks the

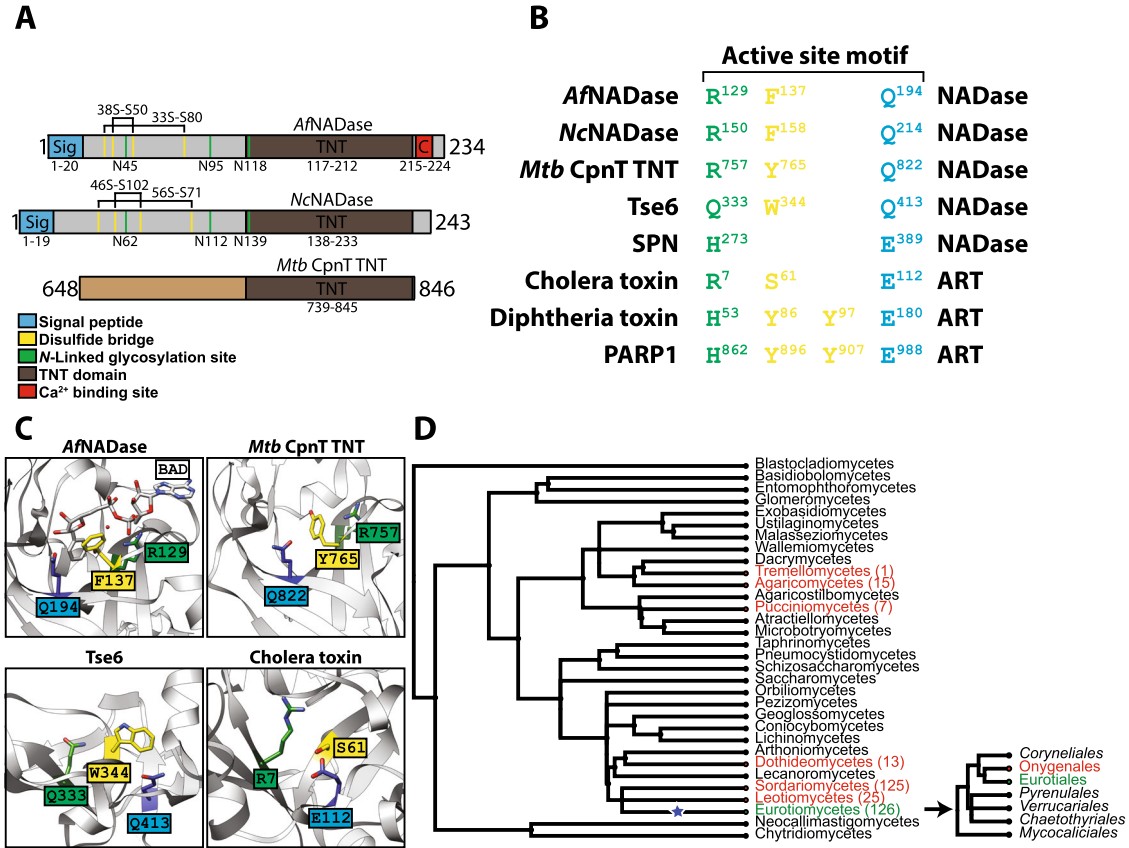

**Fig. 5 *Af*NADase is the founding member of a family of fungal NADases. A** Domain architecture of *Af*NADase, *Nc*NADase and *Mtb* CpnT TNT. **B** Structure-based alignment of active site motifs of NADases and ADP-ribosyltransferases. The residues are found in three distinct regions and are coloured according to their location in the sequence. **C** Comparison of the active sites of *Af*NADase in complex with BAD, *Mtb* CpnT TNT, Tse6 and cholera toxin. Amino acids are coloured based on their position in the active site motif, as in (**B**). **D** Phylogenetic distribution of NADase within the kingdom fungi. Classes with the NADase are coloured red or green, red for classes without the calcium binding site and green for those with the calcium binding site, and the number in parentheses represents the number of species within the class that harbour the NADase. The calcium binding site emerged (blue star) in the Eurotiomycetes, and is only present in the order Eurotiales (inset).

residues involved in Ca²⁺ binding found in *Af*NADase (Fig. 1D). A comprehensive phylogenetic analysis of the distribution of the fungal NADases revealed that they are predominantly present in families and species that are known to be pathogenic suggesting a potential role of the protein in fungal virulence (Fig. 5D). The conidia of *A. fumigatus* are ubiquitously present in the environment and normally cleared from the lungs by the immune system[19]. However, aspergilloses are an increasing problem in animal husbandry and represent a considerable threat for immune-compromised humans[20]. *A. fumigatus* spores harbour an extensive array of virulence factors to establish an infection which could potentially be therapeutic targets[21]. However, the spectrum of antifungal drugs is rather limited, and development of resistance has become a major challenge for the treatment of these infections[22–26]. Interestingly, the gene encoding *Af*NADase is upregulated during conidiation of *A. fumigatus* strains displaying high adherence to pulmonary epithelial cells[27].

In summary, we have discovered a group of fungal surface NADases that have distinct structural and functional features, and have characterised the NADase from *A. fumigatus*. Importantly, no homologues in mammals have been found. Given their functional similarity to known toxins with NADase activity, and predominant presence in pathogenic fungi, we speculate that fungal NADases may convey advantages during interaction with the host or competing microorganisms in the environment.

## Methods

**Fungal strains, media and growth conditions**. *Aspergillus fumigatus* strain CEA17Δ*akuB*[28] was used as parental strain and for generation of the Δ*nadA* knockout and complementation strains. For conidiation, strains were cultivated on *Aspergillus* minimal medium 1.5% (w/v) agar plates at 37 °C (AMM; 70 mM NaNO₃, 11.2 mM KH₂PO₄, 7 mM KCl, 2 mM MgSO₄, (pH 6.5) with 1% (w/v) glucose and 1 µl/ml trace element solution: 18 mM FeSO₄, 171 mM EDTA, 77 mM ZnSO₄, 180 mM H₃BO₃, 25 mM MnCl₂, 6.7 mM CoCl₂, 6.4 mM CuSO₄, 0.9 mM (NH₄)₆Mo₇O₂₄[29,30]. After 3–7 days, conidia were harvested in sterile, ultra-filtrated water using a cell scraper and counted in a Neubauer chamber. For isolation of genomic DNA, *A. fumigatus* was grown in liquid AMM for 24 h.

**Isolation and manipulation of *A. fumigatus* nucleic acid**. Standard techniques for manipulation of DNA were carried out using standard procedures. Chromosomal DNA of *A. fumigatus* was prepared using the Master Pure Yeast DNA Purification Kit (Epicentre Biotechnologies, USA). For Southern blot analysis, DNA fragments were separated on an agarose gel and blotted onto Hybond N⁺ nylon membranes (GE Healthcare Bio-Sciences, Germany). DNA probes were labelled using the DIG labelling mix (Roche Applied Science, Germany) following manufacturers recommendations. Hybridization and detection of DNA-DNA hybrids were performed using DIG Easy Hyb and a CDP-star ready-to-use kit (Roche Applied Science, Germany) following manufacturers recommendations[31].

**Generation of *A. fumigatus* mutant strains**. Deletion of *nadA* was done by using a PCR-based strategy. Upstream and downstream flanking regions of gene *nadA* (AFUA_6G14470) were amplified by PCR using primer pairs 6G14470_5for (GGTCATTGTAAATATCTGGG) and 6G14470_ptrArev (GGCCTGAGTGGC-CATCGAATTCCGCCGTGTAATACTGAGAAG) and 6G14470_ptrAfor (GAGGGCCATCTAGGCCATCAAGCCTTATGGGAAGTG GATCTTG) and 6G14470_3rev (GTAGTGGATAACGAAGATTCG), respectively (Supplementary Table 3). By this reaction overlapping ends to the pyrithiamine

resistance cassette were introduced at the 3'-end of the upstream flanking region and at the 5'-end of the downstream flanking region of the *nadA* gene. The *ptrA* resistance cassette was amplified from plasmid pSK275[32] with primers ptrA-for (GAATTCGATGGCCACTCAGGCC) and ptrA-rev (GCTTGATGG CCTA-GATGGCCTC). All PCR reactions were performed with Phusion Flash Polymerase Master Mix (Thermo Scientific, Germany) according to the manufacturer's recommendations and PCR fragments were purified by gel extraction. The final deletion construct was generated by a three fragment PCR employing primers 6G14470_5for and 6G14470_3rev. The resulting PCR product was purified and used for transformation of *A. fumigatus* protoplasts as by electroporation[33]. Pyrithiamine (1 mg/mL, Merck, Germany) resistant transformants were analyzed for deletion of *nadA* by Southern blot analysis.

To create the complemented strain, the deletion strain *ΔnadA* was complemented in-locus using the same approach and reagents as described above. The *nadA* gene was amplified from genomic DNA of *A. fumigatus* strain CEA17*ΔakuB* using primers 14470_fwd (AGGCGTATCACGAGGCCCTTTCGT CGGTCATTGTAAATATCTGGG) and 14470_rev (CAATAGTGCCACGCT ATTGGGATCACTGGC). The *hph* resistance cassette was amplified from plasmid pUChph[34] using primers hph_fwd (TGATCCCAATAGCGTGGCACTATTGA TCATCC) and hph_rev (GGCCATCGAATTCGCCAGTGTGCTGGAATTC). The *ptrA* resistance cassette was amplified using primers compl_ptrA_fwd (CAGCACACTGGCGAATTCGATGGCCACTCAG) and compl_ptrA_rev (TCACCGTCATCACCGAAACGCGCGAGCTTGATGGCCTAGATGG). The complementation construct consisting of the amplified *nadA* gene (including promoter and terminator), *hph* and *ptrA* resistance cassettes was generated by a multifragment PCR, purified and transformed as described above.

Transformants resistant to hygromycin (150 μg/ml, Roche Applied Science, Germany) were analyzed for complementation of *nadA* by Southern blot analysis.

**Fluorometric determination of NADase activity**. The NADase activity of conidia, purified proteins and medium from cells transiently overexpressing AfNADase or mutants were determined by measuring the increase in fluorescence upon cleavage of the fluorescent NAD analogue nicotinamide 1, εNAD+. Reactions were prepared in 200 μl reaction buffer (50 mM Na-Acetate pH 5.5, 150 mM NaCl, 0.5 mM CaCl2) supplemented with 80 μM εNAD+. The reaction was started by adding conidia, medium from transiently transfected 293 cells or purified AfNADase. The initial reaction rate was monitored by measuring the change in fluorescence over time at 410 nm produced by excitation at 300 nm. The experiments were performed using a Cary Eclipse fluorescence spectrophotometer (Varian)

**Fluorometric determination of NADase activity of *A. fumigatus* at different growth stages**. *A. fumigatus* conidia in Aspergillus minimal medium were incubated in a 96 well plate at 37 °C for the indicated time. The reaction was started by adding εNAD+ to a final concentration of 80 μM and the initial reaction rate was monitored by measuring the change in fluorescence over time at 410 nm triggered by excitation at 310 nm. The experiments were performed using an Infinite M200Pro plate reader (Tecan).

**In gel NADase activity assay**. Samples of *N. crassa* conidia and *A. fumigatus* conidia were prepared by mixing them with 1X non-reducing SDS-PAGE sample buffer (50 mM Tris-HCl pH 6.8, 2 % (v/v) SDS, 10 % (v/v) glycerol, 0,005 % bromophenol blue). In addition, samples of heat-treated *A. fumigatus* conidia were prepared by incubating the spores at 95 °C for 5 min. The samples were run on a 12 % non-reducing SDS polyacrylamide gel at 10 mA, following electrophoresis the gels were washed twice in washing buffer (50 mM Tris-HCl pH 7, 0.5 CaCl2 0.5 % (v/v) NP-40). The gel was developed by incubation for 5 min in developing buffer (50 mM Tris-HCl pH 7, 0.5 CaCl2, 0.5 % (v/v) NP-40, 80 μM εNAD+). The fluorescent bands were visualized with a UV-transilluminator and excised for mass spectrometry analysis.

**Sample preparation for LC-MS/MS analysis**. The bands showing activity in the fluorometric assay (Fig. 1D) were excised from the gel, washed and destained by repetitive alternating incubation in acetonitrile and 50 mM NH4HCO3. Disulfide bonds in the proteins were reduced using 20 mM Tris(2-carboxyethyl)phosphine for 30 min at 55 °C. Reduced cysteine residues were carbamidomethylated using 25 mM iodoacetamide for 30 min at RT before gel pieces were washed and dried. For tryptic digestion, gel pieces were reconstituted with Trypsin/LysC (50 ng/μl, Promega) solution and incubated at 37 °C for 18 h. Peptides were extracted by three steps of sonication (first extraction 1/49/50 trifluoroacetic acid/water/ acetonitrile, second extraction 1/29/70, third extraction 1/9/90). Extracts were pooled, dried and dissolved in 0.05% TFA in 2/98 acetonitrile/water for LC-MS/MS analysis.

**LC-MS/MS analysis**. LC-MS/MS analysis of tryptic peptides was performed on an Ultimate 3000 RSLC nano instrument coupled to a QExactive Plus mass spectrometer (both Thermo Fisher Scientific). Tryptic peptides were trapped for 4 min on an Acclaim Pep Map 100 column (2 cm × 75 μm, 3 μm) at a flow-rate of 5 μL/min. The peptides were then separated on an Acclaim Pep Map column (50 cm × 75 μm, 2 μm) using a binary gradient (A: 0.1% (v/v) formic acid in H2O; B: 0.1% (v/v) formic acid in 90:10 (v/v) ACN/H2O): 0–4 min at 4% B, 10 min at 7% B, 40 min at 10% B, 60 min at 15% B, 80 min at 25% B, 90 min at 30% B, 110 min at 50% B,

115 min at 60% B, 120–125 min at 96% B, 125.1-150 min at 4% B. Positively charged ions were generated by a Nanospray Flex Ion Source (Thermo Fisher Scientific) using a stainless steel emitter with 2.2 kV spray voltage. Ions were measured in data-dependent MS2 (Top10) mode: Precursor ions were scanned at m/z 300-1500 (R: 70,000 FWHM; AGC target: 1·106; max. IT: 120 ms). Fragment ions generated in the HCD cell at 30% normalized collision energy using N2 were scanned (R: 17,500 FWHM; AGC target: 2e5; max. IT: 120 ms) using a dynamic exclusion of 30 s.

**Protein database search**. The MS/MS data were searched against the Uniprot database of *Aspergillus fumigatus* Af293 / *Neosartorya fumigata* Af293 using Proteome Discoverer 2.2 and the algorithms of Mascot 2.4.1, Sequest HT, and MS Amanda 2.0. Two missed cleavages were allowed for tryptic peptides. The precursor mass tolerance was set to 10 ppm and the fragment mass tolerance was set to 0.02 Da. Dynamic modifications were set as oxidation of Met and acetylation of the protein N-terminus. The static modification was set to carbamidomethylation of Cys. One unique rank 1 peptide with a strict target false discovery (FDR) rate of <1% on both peptide and protein level (compared against a reverse decoy database) were required for positive protein hits.

**Determination of substrate specificity by HPLC**. Reactions were prepared in 1 ml reaction buffer (50 mM Tris-HCl pH 8.0, 0.5 mM CaCl2, 150 mM NaCl) supplemented with 200 μM substrate (NAD+, NADH, NADP+, NADPH). After adding 10 ng purified AfNADase and subsequent incubation at 20 °C for 1 h 50 μl of the reaction were analysed by HPLC using a CC 250/3 nucleosil 100-3 C18 HD column (Macherey-Nagel cat.no. 721492 30). Samples were run with a gradient between buffer A (10 mM ammonium acetate pH 7.5, 2 mM TBA-bromide, 10% (v/v) acetonitrile) and buffer B (10 mM ammonium acetate pH 7.5, 2 mM TBA-bromide, 90% (v/v) acetonitrile). The following gradient was used: 0 min: 0.1% B; 1 min: 0.1% B; 7 min: 4% B; 19 min: 22% B; 21 min: 90% B; 23.5 min: 90% B; 26 min: 0.1% B; 35 min: 0.1 % B. The flow rate was set to 0.6 ml/min and the column compartment was kept at 30 °C during the run. The progress of the run was visualised using a UV-VIS detector at the wavelengths of 259 and 340 nm.

**ADPR cyclization and base exchange assayed by HPLC**. Reactions were prepared in 1 ml reaction buffer (Na-PO4 pH 6.8 or Na-Acetate pH 4, 0.5 mM CaCl2) supplemented with 1 mM NA and 100 μM NAD+. After adding 10 ng purified AfNADase and subsequent incubation at 20 °C for 1 h 50 μl of the reaction were analysed by HPLC as described above. *Aplysia californica* ADP-ribosyl cyclase (Sigma-Aldrich, CAS: 135622-82-1) was used as positive control.

**Fluorometric cyclization assay**. The cyclization activity of purified AfNADase was assayed using the NAD+ analogue nicotinamide hypoxanthine dinucleotide (NHD+) which is converted to the fluorescent N7-cIDPR and nicotinamide by ADPR cyclases. Reactions were prepared in 200 μl reaction buffer (50 mM Na-Acetate pH 5.5, 150 mM NaCl, 0.5 mM CaCl2) supplemented with 40 μM NHD+. The reaction was started by addition of 10 ng purified AfNADase and followed by measuring the change in fluorescence over time at 410 nm produced by excitation at 300 nm. The experiments were performed using a Cary Eclipse fluorescence spectrophotometer (Varian).

**Expression and purification of *AfNADase using Sf9 insect cells**. The sequence encoding *A. fumigatus* NADase (gene: AFUA_6G14470) was amplified from cDNA and inserted in-frame with a C-terminal C3 cleavage site and 6xHis-Tag in the pFastBacNKI-ORF-3C-His insect expression vector (NKI Protein Facility, The Netherlands). Briefly, the empty vector, linearized with KpnI, and the amplified insert were separately treated with T4 polymerase in the presence of dTTP to generate single stranded overhangs. The reaction was terminated by adding 25 mM ethylenediaminetetraacetic acid (EDTA) and by heat inactivation at 75 °C for 20 min. The linearized vector and insert were gel purified with the NucleoSpin® Gel and PCR Clean-up kit (Machery-Nagel). The linearized vector and insert were annealed and used for transformation of One Shot™ TOP10 chemically competent *E. coli* cells (ThermoFischer Scientific). Positive clones were identified by colony PCR and plasmids isolated using the NucleoSpin® Plasmid kit (Machery-Nagel), and the sequence of the plasmid was controlled by Sanger sequencing. DH10EMBacY *E. coli* cells (EMBL, Grenoble) were used for bacmid generation following the protocol described by[35]. Sf9 cells (Invitrogen) in solution at a density of 0.5 million cells/ml were transfected with the AfNADase bacmid using the Cellfectin™ II Reagent (ThermoFischer Scientific), and the virus particles were harvested after seven days. Subsequently the recombinant primary virus was amplified to a high-titre viral stock.

For protein expression Sf9 cells were cultivated in Sf-900™ II SFM (Gibco™) medium and infected with high titre viral stocks at a density of 1.5-2 million cells/ml. Three days past growth arrest the cells were pelleted by centrifugation and the medium filtered using a 0.22 μm filter (Merck Millipore). The secreted enzyme was purified by immobilized metal affinity chromatography using a HisTrap™ excel (GE Healthcare) column connected to an ÄKTA pure chromatography system (GE Healthcare) following manufacturers recommendations. The column was washed with washing buffer (50 mM Tris-HCl, pH 8, 300 mM NaCl) and the protein was eluted with elution buffer (50 mM Tris-HCl, pH 8, 300 mM NaCl, 500 mM Imidazole). The purified

protein was concentrated using 10 kDa MWCO Amicon® Ultra Centrifugal filters (Merck Millipore) and further purified by size exclusion chromatography using a Superdex 200 16/60 HiLoad prepgrade column (GE Healthcare) connected to an ÄKTA pure chromatography system (GE Healthcare). The size exclusion purification was performed using a SEC buffer (50 mM Tris-HCl pH 8, 300 mM NaCl, 2 mM TECEP) and typically the yield was 20-25 mg per litre of medium.

**Expression and purification of *Nc*NADase and *Af*NADase using 293 cells.** The sequence encoding *Nc*NADase (gene: NC_026504.1 in N. crassa OR74A) was codon-optimized for mammalian expression, synthesised and inserted into the vector pUC57 by GenScript. The synthesised sequence was flanked by a 5'-end HindIII restriction site and a Kozak sequence, and at the 3'-end by a KpnI restriction site. The resulting pUC57 vector was restriction digested with HindIII and KpnI, prior to ligation into the vector pCMV-Flag5a via the HindIII and KpnI restriction sites. The newly generated plasmid pCMV-*Nc*NADase-Flag was isolated and sequenced. 293 cells were transiently transfected using the Effectene transfection reagent (Qiagen) following manufacturers recommendations, and the cell medium was harvested 96 h post transfection. The medium containing the overexpressed *Nc*NADase was concentrated with 10 kDa MWCO Amicon® Ultra Centrifugal filters and subsequently the enzyme was purified using Anti-FLAG M2 affinity gel (Sigma-Aldrich) following manufacturers recommendations with the following modifications: Triton X-100 and NaCl was added to the medium to the final concentrations of 0,5 % (v/v) and 300 mM, respectively. 40 µL of the affinity gel was added to the medium and incubated for 2 h on a rotating wheel at 4 °C. Subsequently the affinity gel was washed twice with TBS (Tris-HCl pH 7.4, 150 mM NaCl) and then four times with TBS containing 1 M NaCl. The protein was eluted by incubation with 3X FLAG-peptides (5 µg/ul) in TBS for 30 min on a rotating wheel at 4 °C

The sequence encoding *A. fumigatus* NADase (gene: AFUA_6G14470) was amplified from cDNA, using the forward primer 5'-GTTGGATCCCCACCATGAT CTTCACCAAC-3' and the reverse primer 5'-GCATAGAATTCCTAGTGAT GGTGATGGTGATGCTGATTCGGCCCCGGAGTATAC-3'. The resulting PCR amplicon is endowed with a 3'-end BamHI restriction site followed by a Kozak sequence and a 5'-end EcoRI restriction site followed by a sequence encoding a C-terminal hexahistidine tag. The PCR product was restriction digested with the aforementioned restriction enzymes and inserted into the vector pcDNA3.1(+). The resulting plasmid pcDNA3.1(+)-AfNADase-6XHis which encodes a C-terminal his tagged *Af*NADase was isolated and sequenced. Stably transfected 293 cells were generated by transfection using the calcium phosphate precipitation method followed by two rounds of selection in the presence of 550 µg/ml G418. Monoclonal cell lines were adapted to grow in the chemically defined medium Gibco™ FreeStyle™ 293 Expression Medium (Thermo Fisher). The overexpressed protein secreted into the medium was purified as described above for the Sf9 insect cells.

**Generation of *Af*NADase mutants.** Mutants for expression in Sf9 insect cells and 293 cells were generated by mutating the parental plasmids using the Q5® Site-Directed Mutagenesis Kit (New England Biolabs Inc.) following manufacturers recommendations. The primers were designed using the NEB base exchanger web tool (New England Biolabs Inc.) The F137A mutant was generated using the primes 5'-GTATGGCACCGCTCTGGCGCCGC-3' and 5'-TCCGATCCGAAA CGGTCAAG-3'. The Q194A and Q194K was produced with the primers 5'-GATGGGGACGGCTTTCGTGACATATACCAATG-3', 5'-CCTGGCTGCT CAAACCAA-3', 5'-GATGGGGACGAAGTTCGTGACAT-3' and 5'-CCTG GCTGCTCAAACCAAG-3', respectively. The R129A mutant was generated using the primers 5'-GAAGCTTGACGCGTTCGGAGTATGG-3' and 5'-ATGCCAACCGGTAAGGTC-3'. The F130A mutant was produced with the primers 5'-GCTTGACCGTGCGGGATCGGAGTATG-3' and 5'-TTCATGCCAA CCGGTAAG-3'. The calcium site binding mutant (D219A/E220A) was generated with the primers 5'-GAGCGAGTATGCTGCCAAGGTGGAATACTC-3' and 5'-TCATCCAACCGTCGCAAG-3'.

**Crystallization and structure determination.** Crystallization screening for native *Af*NADase was performed in a high throughput manner using a sparse matrix approach utilizing the JCSG + and PACT premier HT screens from Molecular Dimensions. Sitting drop vapour diffusion crystallization trials were set up on Swiss CI SD3 plates at 20 °C and 8 °C against 40 µl reservoir, with ratios of 2:1, 1:1 and 1:2 of protein and precipitant in the drop, respectively. The total drop size was 600 nl and the plates were set up using Mosquito LCP crystallization robot (TTP Labtech). Crystallization was followed with a ROCK IMAGER (Formulatrix) instrument at 20 °C, and manually for the 8 °C plates. The first crystals appeared within a few days and growth continued for up to one-week. Initial crystallization conditions for native AfNADase were 0.1 M sodium acetate pH 5, 0.2 M CaCl₂ and 25 % (w/v) polyethylene glycol (PEG) 6000. Optimized crystals used for data collection were grown at 20 °C using hanging-drop vapour diffusion in conditions containing 0.1 M sodium acetate pH 5, 0.3 M CaCl₂ and 20-25 % (w/v) PEG 8000. Prior to the data collection, crystals were cryoprotected by briefly soaking them in cryo-protectant, consisting of the crystallization solution supplemented with 20 % glycerol (v/v).

Co-crystallization of AfNADase and BAD was performed by adding BAD to a final concentration of 5 mM in the protein solution prior to setting up the crystallization. BAD was a kind gift from John Pascal or synthesised enzymatically from benzamide

riboside. The crystallization condition was the same as for the native protein. The cryo-protectant used for these crystals was supplemented with 10 mM BAD.

EGTA-treated AfNADase was crystallized in 0.04 M potassium phosphate monobasic, 16% PEG 8000 and 20% glycerol. Because of the glycerol in the crystallization condition cryoprotection was not needed but crystals were soaked for 1 min in the crystallization condition supplemented with 10 mM NAD prior to mounting in Litho loops (Molecular Dimensions) and flash freezing in liquid N₂.

Crystals used for single-wavelength anomalous dispersion (SAD) phasing were co-crystallized with 100 mM NaI using the optimized condition for native AfNADase. In addition, the crystals were soaked in cryoprotectant supplemented with 20% glycerol and 500 mM NaI before being mounted in Litho loops and flash frozen in liquid N₂.

Phasing was done at P13, operated by EMBL Hamburg at the PETRA III storage ring (DESY, Hamburg, Germany), utilizing iodide derivatives of *Af*NADase and the SAD method. The data were collected with 6 kEV energy to reach anomalous scattering coefficient of 11 e⁻ (f''). 720° with 0.2° oscillation of data were collected to maximize the multiplicity. The resulting autoprocessed data (XDS and aimless), showing significant anomalous signal, was used for determining the phases with AutoSol[36] and further building of the initial model with AutoBuild[37], both from the PHENIX package[38]. 35 heavy atom sites with partial occupancy were found, and the space group was determined to be P3221. The initial model was then used as a search model for molecular replacement in Phaser[39] with high-resolution native data also collected at P13 using 12.7 keV energy.

Data for the EGTA and NAD treated crystal were collected at the P11 end station operated by DESY at the PETRA III storage ring (DESY, Hamburg, Germany) using 12 keV energy, and the BAD co-crystallization data were collected at the I04-1 at the Diamond Light Source (Ditcot, Great Britain) using 13.5 keV energy.

Data sets were processed using XDS[40] and scaled with Aimless[41]. Crystals structures were refined using Phenix.refine[38] and manual inspection and model building was performed using Coot[42]. Structures were validated with MolProbity[38] during the refinement cycles. Ligand restraints were created using eLBOW[43]. POLDER maps were calculated using the POLDER maps program in the Phenix package[44]. Details for the data collection and refinement are shown in Supplementary Table 1. Structures were illustrated using PyMol and UCSF Chimera. Structures are deposited in the Protein Data Bank under ID: 6YGE, 6YGF and 6YGG.

**Determination of *A. fumigatus* conidia NAD⁺ cleavage products by ¹H NMR.** *A. fumigatus* conidia were incubated at room temperature for 1 h with 500 uM NAD⁺ in NMR reaction buffer (25 mM sodium phosphate pH 5.8, 50 mM NaCl, 5 % (v/v) D₂O) in a final volume of 500 µl. Subsequently the conidia were removed by centrifugation followed by filtration through a 10 kDa MWCO Amicon® Ultra Centrifugal filter (Merck Millipore). The reaction products where identified by NMR and data were collected on a Bruker Ascend 850 MHz instrument fitted with a cryogenically cooled triple resonance 5 mm TCI probe with pulse filed gradients along the z-axis at 23 °C. The reaction products were identified by ¹H NMR using the pulse sequence zgesgppe allowing water suppression using excitation sculpting with pulse field gradients and perfect echo. The spectra were acquired with 16 scans and a recovery delay of 3.9 seconds

**Determination of *Af*NADase kinetics by ¹H NMR.** NMR data were collected on a Bruker Ascend 850 MHz instrument fitted with a cryogenically cooled triple resonance 5 mm TCI probe with pulse filed gradients along the z-axis at 23 °C. The hydrolysis of NAD⁺ and NADP⁺ was measured by ¹H NMR using the pulse sequence zgesgppe allowing water suppression using excitation sculpting with pulse field gradients and perfect echo. The spectra were acquired with eight scans and a recovery delay of 3.9 seconds. The reactions consisted of 500 µM NAD⁺ or NADP⁺ in NMR reaction buffer (25 mM sodium phosphate pH 5.8, 50 mM NaCl, 5 % (v/v) D₂O) in a final volume of 500 µl. Reference spectrums of NAD⁺ and NADP⁺ in the absence of *Af*NADase were acquired. The hydrolysis of NAD⁺ and NADP⁺ were measured by adding *Af*NADase to a final concentration of 0.2 nM, the time of enzyme addition was recorded, and spectra were acquired until the substrate resonances declined to the baseline level. The resulting spectra were manually phased and base line corrected. The resonances were assigned using standard correlation methods. The resonances of interest were integrated using the program Dynamics centre 2.5 (Bruker) and compared to the reference spectra of NAD⁺ and NADP⁺, respectively. The kinetic parameters of *Af*NADase were determined by progress curve analysis as described by Golicnik, M[45] using MatLab.

**Phylogenetic analysis.** To gain further insight into the role and evolutionary origin of *Af*NADase we scanned the NCBI protein database for potential fungal homologues. For this, we obtained the fungal phylogeny from the timetree database at the "class" rank compromising 15 fungal classes[46]. We then performed standard BLAST searches[47]. Our search database contained all NCBI non-redundant protein sequences (nr) and was restricted taxonomically to each of the 15 fungal classes (taxid:4751). We used default parameter settings of blastp but filtered for low complexity regions and set the number of target sequences to 500 and the expected value to 1. Homology to the identified calcium binding motif DE[KV]E from the *A. fumigatus* sequence was used as an indicator for the possibility of calcium binding. We did not identify any homologous sequence of non-Eurotiales species containing this or a highly similar motif, suggesting it is specific to this clade.

## Synthesis of benzamide riboside.

**Fig. 6 Synthesis of 3-(1'-β-D-Ribofuranosyl)benzamide (benzamide riboside).** Synthesis adapted from[48].

**Step 1: 3-(2,3,5-Tri-O-benzyl-1-β-D-ribofuranosyl)benzonitrile.** Under nitrogen atmosphere, a stirred solution of 3-iodobenzonitrile (1.50 g, 6.5 mmol) in anhydrous THF (97 mL) was cooled to −78°C and a solution of isopropylmagnesium chloride (3.70 mL, 2 M in THF) was added thereto. The mixture was stirred at −78 °C for 2 h at −78°C. The above reaction mixture was transferred under nitrogen through a cannula into a solution of 2,3,5-tri-O-benzyl-D-ribono-1,4-lactone (2.73 g, 6.5 mmol) in anhydrous THF (18 mL) stirred at −78 °C. The stirred reaction mixture was allowed to react at −78 °C for 2 h. It was allowed to reach ambient temperature (25 °C) and left at this temperature and at stirring overnight to give a pink solution. Next day, a saturated aqueous solution of sodium bicarbonate (60 mL) was added, and the mixture was extracted with ether (ca. 300 mL). The organic layer was separated, washed with brine, separated again, dried over $Na_2SO_4$, filtered, and concentrated under vacuum to give yellow oil (3.41 g). The crude product was dissolved in toluene and volatiles were evaporated (repeated 3 times) to remove any possible traces of water. The residue was diluted with anhydrous $CH_2Cl_2$ (16 mL), and the solution was cooled under nitrogen atmosphere to −78 °C. Boron trifluoride etherate (1.65 mL, 13.1 mmol) was added from a syringe by drops (over ca. 5 min period) to the solution, followed by addition of triethylsilane (2.1 mL, 13.1 mmol). The reaction solution was stirred at −78 °C for 1 h. The solution was allowed to reach ambient temperature (25°C) and stirred overnight. The reaction was quenched with a saturated aqueous solution of $NaHCO_3$ and the reaction mixture was extracted with $CH_2Cl_2$. Organic layer was separated, dried over $Na_2SO_4$, filtered, and evaporated under reduced pressure to give pink-yellow oil (2.98 g). The crude product was twice purified by MPLC column chromatography (100 g SNAP Ultra $SiO_2$ Biotage column, $l = 15.5$ cm; $d = 3.7$ cm) with UV detection (254 and 280 nm) with elution in stepwise gradient mode from hexanes to hexanes/EtOAc mixtures (containing from 15% to 25% of EtOAc) to give 3-(2,3,5-tri-O-benzyl-1-β-D-ribofuranosyl)benzonitrile (1.708 g, 51%) as a colourless oil. [1]H NMR (δ, CDCl₃, 400 MHz): 3.58 and 3.65 (AB part of ABX system, $J_{AB}$ = 10.3 Hz, $J_{AX}$ = 4.0 Hz, $J_{BX}$ = 3.6 Hz, 2H, H5'ₐ and H5'ᵦ), 3.73 (dd, J = 7.7 Hz, 5.1 Hz, 1H, H2'), 4.01 (dd, J = 5.1 Hz, 3.2 Hz, 1H, H3'), 4.34 − 4.37 (m, 2H, H4' and 1/2 CH₂ from Bn), 4.51−4.61 (m, 5H, 2×CH₂ and 1/2 CH₂ from Bn), 4.97 (d, J = 7.4 Hz, 1H, H1'), 7.12−7.15 (m, 2H), 7.23−7.38 (m, 14H), 7.53 (d, J = 7.7 Hz, 1H), 7.59 (d, J = 7.7 Hz, 1H), 7.66 (s, 1H). [13]C NMR (δ, CDCl₃, 100 MHz): 70.36 (C5'), 72.01 (CH₂Ph), 72.52 (CH₂Ph), 73.57 (CH₂Ph), 77.32 (C3'), 81.13 (C1'), 82.36 (C4'), 83.81 (C2'), 112.35 (C−CN), 118.87 (CN), 127.72, 127.82, 127.90, 127.92, 127.95, 128.13, 128.38, 128.46, 128.51, 128.93, 129.71, 130.82, 131.28, 137.31, 137.71, 137.82, 142.12.

**Step 2: 3-(2,3,5-Tri-O-benzyl-1-β-D-ribofuranosyl)benzamide.** To a stirred solution of 3-(2,3,5-tri-O-benzyl-1-β-D-ribofuranosyl)benzonitrile (1.50 g, 3.0 mmol) in acetone (8 mL), hydrogen peroxide−urea complex CO (NH₂)₂•H₂O₂ (1.12 g, 12.0 mmol) was added, followed by addition of water (3 mL) and potassium carbonate (0.041 g, 3.0 mmol). The reaction mixture was stirred at room temperature (25°C) overnight. Next day, additional amounts of CO(NH₂)₂•H₂O₂ (1.00 g), potassium carbonate (0.04 g) and acetone (2 mL) were added, and the reaction was allowed to proceed under the same conditions for 3 days. After this time, the reaction mixture transformed into a clear colourless solution and TLC indicated completion of the reaction. The reaction mixture was diluted with water, dichloromethane was added. Organic material was extracted with EtOAc. Organic phase was separated, dried over $Na_2SO_4$, filtered, and evaporated under reduced pressure to give colourless viscous substance. Dichloromethane was added to the product and evaporated to remove traces of EtOAc. The procedure was repeated to give sticky product gradually transformed into a white crystalline substance (1.45 g, 93%). [1]H NMR (δ, CDCl₃, 400 MHz): 3.65 and 3.75 (AB part of ABX system, $J_{AB}$ = 10.5 Hz, $J_{AX}$ = 3.3 Hz, $J_{BX}$ = 3.0 Hz, 2H, H5'ₐ and H5'ᵦ), 3.82 (t, J = 5.7 Hz, 1H, H3'), 4.06 (t, J = 4.4 Hz, 1H, H4'), 4.35 −4.39 (m, 1H, H4'), 4.44−4.65 (m, 6H, 3×CH₂ from Bn), 5.06 (d, J = 6.3 Hz, 1H, H1'), 5.66 − 5.92 (br m, 2H, NH₂), 7.18−7.20 (m, 2H), 7.25−7.38 (m, 14H), 7.51 (d, J = 7.6 Hz, 1H), 7.76−7.79 (m, 2H). [13]C NMR (δ, CDCl₃, 100 MHz): 70.32 (C5'), 72.05 (CH₂Ph), 72.36 (CH₂Ph), 73.54 (CH₂Ph), 77.17 (C3'), 81.85 (C1'), 82.08 (C4'), 83.68 (C2'), 124.60, 127.31, 127.61, 127.82, 127.85, 127.89, 128.05, 128.37, 128.43, 128.51, 128.73, 130.21, 132.83, 137.53, 137.79, 137.98, 141.09, 169.49 (C = O). MS, *m/z*: 524.14 [M + H]⁺ (Fig. 6).

**Step 3: 3-(1-β-D-Ribofuranosyl)benzamide.** In a 100 mL flask, under nitrogen atmosphere, 3-(2,3,5-tri-O-benzyl-1-β-D-ribofuranosyl)benzamide (1.30 g, 2.5 mmol) was dissolved in anhydrous dichloromethane (50 mL) at stirring. The solution was cooled to −78°C and a 1 N solution of BBr₃ in CH₂Cl₂ (10 mL; 10 mmol; 4 equiv.) was slowly added dropwise from a syringe. The reaction mixture was stirred at −78°C for 1 h 20 min. Cooling bath was removed and the reaction mixture was allowed to reach room temperature and left at stirring overnight at room temperature. Next day, methanol (20 mL) and dichloromethane (20 mL) were added, and the reaction mixture was evaporated to dryness under reduced pressure. The residue was chromatographed on a silica gel column, starting with dichloromethane as eluent and subsequently using mixtures of dichloromethane/ MeOH with gradient from 100/10 to 100/30. Appropriate fractions containing 3-(1-β-D-ribofuranosyl)benzamide were collected, combined and evaporated to give almost colourless viscous substance that was dried under vacuum to give white solid foam (0.50 g). [1]H NMR showed that compound contained ca. 1/2 molecule of MeOH per 1 molecule of 3-(1-β-D-ribofuranosyl)benzamide. Yield: 68%. [1]H NMR (δ, DMSO-d₆, 400 MHz): 3.17 (s, 1.5H, MeOH), 3.53 − 3.61 (m, AB part of ABX system, $J_{AB}$ = 11.8 Hz, $J_{AX}$ = 4.5 Hz, $J_{BX}$ = 4.7 Hz, 2H, H5'ₐ and H5'ᵦ), 3.71 (t, J = 6.3 Hz, 1H, H2'), 3.81 − 3.84 (m, 1H, H3'), 3.90 (t, J = 4.3 Hz, 1H, H4'), 3.98−4.25 (br s, 3H, OH overlapped with water in DMSO-d₆), 4.60 (d, J = 7.1 Hz, 1H, H1'), 7.33 (br s, 1H, 1/2 NH₂), 7.40 (t, J = 7.8 Hz, 1H), 7.55 (d, J = 7.5 Hz, 1H), 7.77 (d, J = 7.5 Hz, 1H), 7.86 (s, 1H), 7.33 (br s, 1H, 1/2 NH₂). [13]C NMR (δ, DMSO-d₆, 100 MHz): 49.03 (MeOH), 62.47 (C5'), 71.83 (C3'), 77.94 (C1'), 83.26 (C4'), 85.68 (C2'), 125.89, 126.82, 128.35, 129.55, 134.55, 141.91, 168.42 (C = O). MS, *m/z*: 254.16 [M + H]⁺, 276.13 [M + Na]⁺, 295.16 [M + H + CH₃CN]⁺, 317.13 [M + Na+CH₃CN]⁺.

**Reporting summary.** Further information on research design is available in the Nature Research Reporting Summary linked to this article.

## Data availability
The sequence data for *Aspergillus fumigatus* and *Neurospora crassa* were deposited in the NCBI GenBank with the accession codes MT276230 (https://www.ncbi.nlm.nih.gov/nuccore/MT276230) and MT316195 (https://www.ncbi.nlm.nih.gov/nuccore/MT316195), respectively. Crystallography atomic coordinates and structure factors were deposited in the Protein Data Bank (PDB) with accession codes 6YGE (https://www.rcsb.org/structure/6YGE), 6YGF (https://www.rcsb.org/structure/6YGF) and 6YGG (https://www.rcsb.org/structure/6YGG). Source data are provided with the paper and any further information will be provided upon request. Source data are provided with this paper.

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

## Acknowledgements

This study was supported by the Research Council of Norway (RCN, No. 302314 to MZ). We made use of the Facility for Biophysics, Structural Biology and Screening at the University of Bergen (BiSS), which has received funding from the RCN through the NORCRYST (grant number 245828) consortium. This work was also supported by the RCN through the Norwegian NMR Platform, NNP (226244/F50). Synchrotron MX data was collected at various beamlines, and therefore we would like to acknowledge DESY (Hamburg, Germany), a member of the Helmholtz Association HGF, for the provision of experimental facilities. Parts of this research were carried out at PETRA III and we would like to thank beamline scientist Eva Crosas for assistance in using beamline P11. We would also like to acknowledge P13 operated by EMBL Hamburg at the PETRA III storage ring and to thank Johanna Hakanpää for the assistance in using the beamline. We are also grateful for Diamond Light Source for beamtime, and the staff of beamlines I04 for their assistance. We thank Thomas Krüger for LC-MS/MS measurements. Work in the laboratory of A.A.B of O.K was supported by the Deutsche Forschungsgemeinschaft Collaborative Research Centre/Transregio 124 FungiNet (projects A1 and Z2).

## Author contributions

M.Z. conceived the project with input from Ø.S. Ø.S. and J.P.K. solved all the structures. A.P. and T.H. cultivated and harvested conidia from *A. fumigatus*. T.H., O.K and A.A.B. designed fungal experiments and interpreted the data. A.P. and T.H. generated the *nadA* KO and complementation strains. A.P. and T.H. prepared samples for the MS experiments. H.M.H. and R.H.S. performed all wet lab work on *Neurospora crassa*. L.J.S. cloned the *Aspergilus fumigatus* NADase and ran the HPLC samples. T.I.G. conducted the phylogenetic analysis. M.M. and M.V.H. synthesised benzamide riboside. Ø.S conducted all other biochemical experiments with the assistance of A.P. Ø.S. and M.Z wrote and revised the manuscript with contributions from all authors.

## Competing interests

The authors declare no competing interests.
