## [Peer Review File · Nature Communications]

Reviewer comments, first round:

Reviewer #1 (Remarks to the Author):

In their manuscript "Conidial surface NADase – a new protein family in pathogenic fungi" Strømmland and colleagues identify and characterise an enigmatic fungal NADase. The paper represents a refreshing contribution to the NAD metabolism field; it provides novel insights into the NADases in microbial warfare (including the first example of these strategies in the realm of fungi) and excellent mechanistic characterisation. Given the emergence of antimicrobial resistance in fungi these new data are of great value.

Specific comments

1) N-linked glycosylation

The authors show that both the AfNADase and NcNADase are glycosylated via N-linkages. However, their data show a clear dependence on the expression background in terms of molecular mass, thermal stability and enzymatic activity. These observations should be discussed further including their potential impact on the conclusions.

2) Ca-dependence and independence of fungal NADases

The authors show that the AfNADase is Ca-dependent and predict through modelling and sequence comparison that the NcNADase is Ca-independent. The authors should firm up this finding with experimental data showing that EGTA and Ca have no effect on the catalytic activity of NcNADase.

3) Phylogenetic

The authors suggest that the fungal NADases identified in this manuscript are a new protein family. However, it would be good to state that they may still represent a highly diverged member of the bacterial TNT domain family.

The authors should also comment on the emergence of the Ca-dependence, e.g. is it a result of gene duplication as would be indicated by species having more than one NADase (one dependent and one independent), or are species carrying the Ca-dependent NADase characterised growth requirements or by the colonization of specific host environments in which a Ca-dependence may present an advantage.

4) Crystallography

The authors demonstrate the presence of their ligands by presenting 2mFo-DFc maps, however, these represent modelled atoms and as such are not a proof of the presence of these atoms. It would be better if the authors showed the OMIT or POLDER maps (or similar) to prove the presence of the discussed ligands.

Extended data table 2 shows that the B-factors of the modelled ligands are high in comparison with macromolecule and solvent. The authors should comment on this and approaches used to solve it (e.g. have they adjusted occupancies in their models and did this affect the B-factors or do they see certain parts more disordered which may be due to loose interactions with the macromolecule).

The authors should correct the presentation of extended data table 2: The abbreviations used should be consistent within their manuscript (e.g. RMSD vs RMS) as well as with the usual representation of the different parameters (e.g. R-work vs. Rwork). This extends also to the units given as e.g. R values are usually given in %. Furthermore, the authors should include "Rotamer (favoured)" as they only give "Rotamer (outliers)" and thus a critical parameter is missing.

5) Catalytic activity / mechanisms

The authors should consider to further analyse their structural data (and, if available, other structures in the PDB) and include a section discussing why the presented fungal NADases are

exclusive hydrolases and cannot catalyse the cyclase reaction found in other NADases.

Reviewer #2 (Remarks to the Author):

This manuscript by Stromland et al. describes the discovery, biochemical and structural characterization of the NAD glycohydrolase of *Aspergillus fumigatus*. This NADase is found on the surface of airborne spores and has remarkable similarity to the TNT, the secreted NADase of *Mycobacterium tuberculosis*. The strengths of this manuscript are the structures of the dimeric enzyme itself and in complex with a non-hydrolyzable NAD analog. This allowed the authors to propose a plausible mechanism of NAD hydrolysis by AfNADase, which may apply to all TNT-like NADases. The manuscript is well written and the experiments are well done.

Critique

Major issues:

1. The major problem of the manuscript is the misclassification of the AfNADase and similar NADases of other fungi as a "hitherto unknown protein family". A simple Blast search with TNT includes many fungal NADases including the NADases of *A. fumigatus* (AFUA_6G14470) and *N. crassa* (NCU07948). Indeed, both proteins are annotated as members of the TNT family (PF14021) in the Pfam database. This is not only based on the extensive sequence similarity as shown in extended figure 7, but is strongly supported by the data presented in the study. The active sites of the TNT and of the NADases of *A. fumigatus* and *N. crassa* are identical. The core of the protein, a seven-stranded central beta-sheet and the flanking α -helices, is structurally identical to that of TNT with an RMSD of 0.8 Å. However, there are notable differences including two disulfide bridges and dimer formation. These features appear to be shared among the fungal NADases, indicating that they form a subfamily of the TNT family.

This issue can be easily fixed by editing the summary, introduction and interpretation (e.g. lines 275-277).

2. If simple Blast/Pfam searches identify AfNADase, there is no justification for an extensive description of the experimental discovery of the AfNADase (pages 2-3). This section should be shortened drastically and the figures 1D, 1E should be moved to the supplement.

3. The crystal structures have a high resolution, but are poorly refined. The R_{free} are 2-4% higher than where they should be at 1.6 Å. The number of Ramachandran outliers is larger than 1%, but should be smaller than 0.1%, possibly zero. The authors have to deposit more accurate structural models. The resolution enables them to do so.

Other comments/issues:

Why is there no antitoxin? Probably the presence of the NADase on the spore surface poses no risk for the viability of the spore. This is in contrast to the NADases in bacteria. This should be discussed.

l. 118: The "exchange of the nicotinamide moiety by nicotinic acid" is rather a hydrolysis of nicotinamide.

l. 131: the unit of turnover rates is per second (small "s").

Reviewer #3 (Remarks to the Author):

In this study, the authors have identified NADase activity associated with the conidial surface of *Aspergillus fumigatus*, an airborne fungal pathogen, obtained it in its recombinant form by expressing the responsible gene in two different systems (insect Sf9 cells and human 293 cells) and deduced its crystal structure. They took the advantage of several biophysical techniques to study the activity as well as the structure of NADase. The structural elucidation of AfNADase is well

performed and detailed. Based on the activity observed and the sequence similarity with other microbial toxins with NADase activity, the authors propose conidial NADase as the possible virulence factor mediating/facilitating fungal infection. Their speculation is based on the depletion of NAD⁺ leading to cell death, as fungal NADase lack additional cyclase activity. However, the experimental evidence proving their speculation is lacking in this study, and they do not discuss about alternative role played by ADPR/Nam, the products of fungal NADase activity.

Major comments:

1. The authors have used CEA17 Δ akuB as the parental strain to generate Δ nadA mutant. Did they check NADase activity with other clinical isolates? Moreover, the authors have grown *A. fumigatus* on minimal medium to obtain conidia; nevertheless, is conidial associated NADase activity universally present whatever the culture medium used for conidiation? Because, fungal surface architecture as well as its secretome is variable with the growth medium/culture conditions.
2. What about the expression of NADA at different growth stages/morphotypes? Was it only conidia associated, or secreted/released into the culture medium during growth?
3. Why the NADase activity is stable even at 95oC, is it because of the short inactivation time of 5 min? If *A. fumigatus* NADase functions as a dimer, then this heat treatment should destroy the quaternary structure of NADase, as the differential scanning calorimetry in Extended data figure 1 indicates the T_m of AfNADase to be around 78oC.
4. 1H-NMR: the reported chemical shifts (ppm) for NAD H2, H4 and H6 are around 9.3, 8.8 and 9.1, respectively, which looks shifted specifically for H4 in the present study. How do the authors explain this discrepancy in the chemical shift?
5. Even in the presence of EGTA, there is significant NADase activity (Figure 2B), which means the activity is only partially Ca²⁺ dependent or is it because only one concentration of EGTA was tried for the assay? Moreover, even after the mutation of D219 and E220, although there was seven-fold decrease, there was still significant activity compared to control. Probably these mutations may alter the structure leading to the observed decrease in the activity than the defective Ca²⁺ binding resulting in the decreased activity.
6. Why the conserved residues R129, R148 and F130, if important, were not attempted to mutate to establish their importance?
7. Macrophages and neutrophils being the major innate immune cells against *A. fumigatus*, how differentially the CEA17 Δ akuB and Δ nadA mutant conidia will interact with them? What will be their virulence in in vivo model?
8. ADP-ribose, upon polymerization, is also known to regulate many physiological processes. ADPR-polymer is also a potent activator of proinflammatory cytokines. Therefore, how do the authors rule out the possibility of fungal NADase function in providing the substrate ADPR for host ADPR-polymerases for host-benefits?

Minor points:

1. Title: The authors have identified only one NADase in *Aspergillus fumigatus*, a pathogenic fungus; but the other NADase is from *Neurospora crassa*, which is more of a model fungus. Therefore, a general title, 'a new protein family in pathogenic fungi', is not appropriate.
2. Line 18: Expand NAD in the abstract.
3. Line 19: how do the pathogens inject NADases; is 'release' a better word? Or, anything known about the mechanism of injecting NADases by pathogens?
4. Lines 21-22: the authors have found only one NADase on the conidial surface, therefore 'NADases' and 'these' may not be appropriate.
5. Line 42: expand TIR domain.
6. Lines 46-48: the statements are contradictory; the authors mention no NADase has been identified in fungi, but then they mention that NADase activity has been reported in *Neurospora crassa*. Indeed, Ref. 13 is about purification and characterization of NADase (NAD⁺-glycohydrolase) from the conidia of *Neurospora crassa*.
7. Figure 1D: *A. californica* cyclase has been described in the methods as a positive control for HPLC, but not in this figure legend.
8. Figure 1F: NADase activity of the complementation strain is lacking.
9. Methods, fungal strains, media and growth conditions: replace 'formation of conidia' with 'conidiation'.
10. In the methods section, the details of in gel NADase activity is not clear enough; *N. crassa* and *A. fumigatus* conidia (both heated and non-heated) were mixed with non-reducing SDS-PAGE

sample buffer for how long, to see NADase activity?

11. Substrate specificity by HPLC: is there specific reason for performing the reaction at 20°C? Provide a reference for visualization of the compound running through the HPLC column at 259 nm and 340 nm.

12. Why there is discrepancies in the molecular weight of AfNADase expressed in HEK293 cells by size exclusion chromatography and through partition coefficient?

13. One of the Ca²⁺ binding residue has been mentioned as S219 in the text, but in the Figure 3 as S216; which one is correct?

14. Why there are two (may be three) bands after PNGase treatment of NcNADase (Extended data figure 5)?

15. Whether the tuberculosis necrotizing toxin domain was absent in the NcNADase?

16. Is NcNADase activity Ca²⁺ independent?

17. What is the importance of NcNADase, as it is a saprophyte.

18. Af and Nc should be in italics before NADase, as they represent fungal species.

19. Is there any detail about the glycosylation pattern of the recombinant proteins? Are they not important for NADase activity?

20. Whether the NADase activity of CpnT is Ca²⁺ dependent?

Gaaino PDF Trial
www.gaaino.com

Response to the reviewers' comments

We would like to thank the reviewers for their insightful comments and constructive criticisms. Based on their suggestions, we have conducted various additional experiments to strengthen the conclusions and to clarify the points raised.

As detailed below, to present the new data, we have re-arranged the main figures (now 5) and added supplemental figures (or new panels to the previous versions). We believe that the new data have consolidated our observations and further support the conclusions.

Point by point reply to the individual comments

Reviewer #1 (Remarks to the Author)

In their manuscript "Conidial surface NADase – a new protein family in pathogenic fungi" Strømland and colleagues identify and characterise an enigmatic fungal NADase. The paper represents a refreshing contribution to the NAD metabolism field; it provides novel insights into the NADases in microbial warfare (including the first example of these strategies in the realm of fungi) and excellent mechanistic characterisation. Given the emergence of antimicrobial resistance in fungi these new data are of great value.

We thank the reviewer for the favorable comments that highlight the novelty and importance of our observations and conclusions.

Specific comments

1) N-linked glycosylation

The authors show that both the AfNADase and NcNADase are glycosylated via N-linkages. However, their data show a clear dependence on the expression background in terms of molecular mass, thermal stability and enzymatic activity. These observations should be discussed further including their potential impact on the conclusions.

Indeed, there is a difference in apparent molecular mass (as estimated by gel electrophoresis) depending on the expression system (Fig. S2). It is known that the complexity of protein glycosylation is higher in mammalian cells (here 293) compared to insect cells (Sf9). Accordingly, the estimated MW of the recombinant protein expressed in 293 cells is slightly higher than that from Sf9 cells. We now mention this difference in the text. Interestingly, the migration of the NADases from conidia of *A. fumigatus* and *N. crassa* would indicate a molecular mass of ~50 kDa and consequently a higher complexity of the glycosylation. However, the sample is far more complex (conidia vs. purified recombinant proteins) and the electrophoresis conditions differ (no reducing agent present for the activity gels). Therefore, we are reluctant to compare these observations and draw conclusions regarding the extent of glycosylation. However, we agree that the composition of the glycosylation may have effects on the catalytic activity. The thermal stability was only assessed for the protein purified from insect cells. Likewise, enzyme kinetics were conducted only with the protein expressed in Sf9 cells, because of higher yields and purity.

2) Ca-dependence and independence of fungal NADases

The authors show that the AfNADase is Ca-dependent and predict through modelling and sequence comparison that the NcNADase is Ca-independent. The authors should firm up this finding with experimental data showing that EGTA and Ca have no effect on the catalytic activity of NcNADase.

We agree with the reviewer that this supposition is amenable to experimental verification. As shown in the new supplemental Fig. 2I, the NADase from *N. crassa* is indeed Ca^{2+} -independent. Neither addition of Ca^{2+} or EGTA had a measurable effect on enzyme activity. As pointed out by the reviewer, these observations strengthen our conclusions regarding the role of Ca^{2+} as a regulator of the enzyme from *A. fumigatus*.

3) Phylogenetic

The authors suggest that the fungal NADases identified in this manuscript are a new protein family. However, it would be good to state that they may still represent a highly diverged member of the bacterial TNT domain family.

The reviewer is correct, the fungal NADases share the TNT domain with the bacterial enzymes, even though this was not obvious from the initial sequence alignments (Supplemental Fig. 8). The 3D structures indicate that the TNT fold is well preserved, while the primary structures have diverged. As further elaborated in the response to reviewer 2, the assignment of proteins to specific families based on a single domain is often ambiguous. The fact that no fungal NADases have been identified so far, even though they seem to be widely represented in this kingdom, as well as their distinct structural properties beyond the TNT, in our view, classify them as a protein family of their own. Nevertheless, as suggested by the reviewer, we now make more extensive reference to the commonalities and differences between the fungal and bacterial proteins.

The authors should also comment on the emergence of the Ca-dependence, e.g. is it a result of gene duplication as would be indicated by species having more than one NADase (one dependent and one independent), or are species carrying the Ca-dependent NADase characterised growth requirements or by the colonization of specific host / environments in which a Ca-dependence may present an advantage.

We are grateful for pointing out to us that there appear to be some fungal species (such as *Fusarium spp.*) that harbor more than one gene putatively encoding an NADase. For this particular example, both genes appear to lack the calcium binding site. However, to appropriately address the possibility raised by the reviewer would require a more extensive bioinformatics analysis along with experimental verification. As may be inferred from the manuscript, we are ourselves puzzled by the existence and function of the calcium site. At this point, we can only speculate that it might represent a kind of “immunity factor” to minimize endogenous NAD cleavage. We now indicate this thought in the text. Clearly, it will be very important and interesting to establish the origin and purpose of the calcium binding site, in particular, as it seems to be present only in a rather limited number of species.

4) Crystallography

The authors demonstrate the presence of their ligands by presenting 2mFo-DFc maps, however, these represent modelled atoms and as such are not a proof of the presence of these atoms. It would be better if the authors showed the OMIT or POLDER maps (or similar) to prove the presence of the discussed ligands.

Extended data table 2 shows that the B-factors of the modelled ligands are high in comparison with macromolecule and solvent. The authors should comment on this and approaches used to solve it (e.g. have they adjusted occupancies in their models and did this affect the B-factors or do they see certain part more disorder which may be due to lose interactions with the macromolecule).

The authors should correct the presentation of extended data table 2: The abbreviations used should be consistent within their manuscript (e.g. RMSD vs RMS) as well as with the usual representation of the different parameters (e.g. R-work vs. Rwork). This extends also to the units given as e.g. R values are usually given in %. Furthermore, the authors should include “Rotamer (favoured)” as they only give “Rotamer (outliers)” and thus a critical parameter is missing.

We thank the reviewer for the valuable and important criticisms. In fact, they guided us to discover a mistake in the original version of the manuscript. The individual changes and adjustments we have made based on these comments are outlined below.

POLDER maps have been calculated for ADPR-Nam and BAD using Phenix. The figure with fo-fc POLDER maps contoured at 3σ has been added to the manuscript to unambiguously show the presence of the ligands.

B-Factor values in the table labelled as “ligand” also include carbohydrates. The individual B-factors for the ligands have been added to the table. For Nam and BAD the B-factors are well in line with the B-factors for surrounding amino acids and solvent, however, for ADPR the B-factors are elevated. This might be due to the lack of strong interaction for both the ribose and adenosine moieties. Indications of flexibility can be observed for ADPR, and the adenosine moiety of BAD.

We adjusted both occupancies and tried multiple conformations during the refinement process. Occupancies lower than 1 resulted in positive difference density for the phosphate part of ADPR. Selecting only one conformation with an occupancy of 1 gave the best results.

The data tables have been corrected according to the reviewer’s recommendations. Unfortunately, a mistake was made when the table was generated, and the label “Rotamer (outliers)” should have been “Ramachandran outliers”. The actual values for Rotamer allowed (not shown in the table) are between 98.3-99.4 % for all the structures.

5) Catalytic activity / mechanisms

The authors should consider to further analyse their structural data (and, if available, other structures in the pdb) and include a section discussing why the presented fungal NADases are exclusive hydrolases and cannot catalyse the cyclase reaction found in other NADases.

The cyclase reaction requires the formation of a ribooxocarbenium ion reaction intermediate; this intermediate is usually stabilized by an invariant glutamic acid, e.g. glutamic acid 218 of human CD38. However, in fungal NADases a glutamine, glutamine 194 in *Af*NADase, is found in the corresponding position that precludes the formation of a oxocarbenium ion reaction intermediate which is required for the formation of cADPR. A discussion on this critical aspect has now been included in the manuscript.

Reviewer #2 (Remarks to the Author)

This manuscript by Stromland et al. describes the discovery, biochemical and structural characterization of the NAD glycohydrolase of *Aspergillus fumigatus*. This NADase is found on the surface of airborne spores and has remarkable similarity to the secreted NADase of *Mycobacterium tuberculosis*. The strength of this manuscript are the structures of the dimeric enzyme itself and in complex with a non-hydrolyzable NAD analog. This allowed the authors to propose a plausible mechanism of NAD hydrolysis by *Af*NADase, which may apply to all TNT-like NADases. The manuscript is well written and the experiments are well done.

We very much appreciate the favorable overall assessment of our study by the reviewer.

Critique

Major issues:

1. The major problem of the manuscript is the misclassification of the AfNADase and similar NADases of other fungi as a “hitherto unknown protein family”. A simple Blast search with TNT includes many fungal NADases including the NADases of *A. fumigatus* (AFUA_6G14470) and *N. crassa* (NCU07948). Indeed, both proteins are annotated as members of the TNT family (PF14021) in the Pfam database. This is not only based on the extensive sequence similarity as shown in extended figure 7, but is strongly supported by the data presented in study. The active sites of the TNT and of the NADases of *A. fumigatus* and *N. crassa* are identical. The core of the protein, a seven-stranded central beta-sheet and the flanking α -helices, is structurally identical to that of TNT with an RMSD of 0.8 Å. However, there are notable differences including two disulfide bridges and dimer formation. These features appear to be shared among the fungal NADases, indicating that they form a subfamily of the TNT family. This issue can be easily fixed by editing the summary, introduction and interpretation (e.g. lines 275-277).

We agree with the reviewer that a major structural component of the fungal NADases is the TNT domain, which we experimentally verified to be the catalytic core. However, as described in the manuscript, in our hands, a simple BLAST search did not return the *M. tuberculosis* (or other bacterial) TNT protein when searching with the *A. fumigatus* sequence. We then discovered the presence of the TNT domain in the fungal NADases based on a Pfam search. Eventually, with the help of the 3D structural information, we could generate a more appropriate sequence alignment of the fungal and bacterial TNT domains (Fig. S8). Despite their considerable sequence differences, we also agree that the fold is well preserved.

When it comes to the assignment of protein families, this is not unambiguous. See, for example, the definition used at the renowned EBI: <https://www.ebi.ac.uk/training/online/course/introduction-protein-classification-ebi/protein-classification/what-are-protein-families>. By their classification, “subfamily” refers to a small group of proteins, which seems inappropriate for the very large and wide-spread number of fungal NADases. Moreover, as indicated by the entries in the Pfam database, many proteins containing a TNT domain also contain other domains. Which domain should determine the protein family they belong to? The fungal NADases have at least one other distinct domain (the “thumb” domain) with unique structural properties, not present in the bacterial TNT proteins. Namely, the thumb domain contains characteristic disulfide bridges and glycosylation sites. The function of this domain is unknown, but could be important for fungal physiology. It should also be noted that the fungal NADases assemble as functional dimers, a property not known for bacterial TNT proteins. Perhaps, the most distinct feature of the fungal NADases is their presence on the surface of conidia and hyphae, as opposed to being complexed inside the cells by an immunity factor, which is the case for the *M. tuberculosis* TNT. For these reasons, we believe it is most appropriate to classify the fungal NADases as a protein family of their own. We do, however, see the point raised by the reviewer, namely, that the presence of the TNT domain should be highlighted appropriately, and we have now done so in all parts of the text, including Abstract and Introduction.

2. If simple Blast/Pfam searches identify AfNADase, there is no justification for an extensive description of the experimental discovery of the AfNADase (pages 2-3). This section should be shortened drastically and the figures 1D, 1E should be moved to the supplement.

With all due respect, we believe that the *functional* discovery of the fungal NADases is a major accomplishment of the present study. Apparently, things are not as easy as the reviewer suggests. In a recent article from the Niederweis group (DOI: [10.1074/jbc.RA118.005832](https://doi.org/10.1074/jbc.RA118.005832)), sequence alignments were presented (in Fig. 1C of that paper) in which putative sequences of *N. crassa* and *A. fumigatus* TNT proteins were presented. However, the sequences shown are not present in the enzymes we identified. Moreover, in that paper, the sequence stretches shown for these two organisms are identical. When scrutinizing this sequence, we found that it most likely originates from a bacterial, rather than fungal species. Perhaps, this bacterial sequence is a contamination of the database the authors used to retrieve fungal sequences. We are reluctant to raise this point in our manuscript. However, given this very recent erroneous indication of fungal NADase sequences in the literature, we believe it is even more important to demonstrate that we identified the fungal NADase genes based on the functional property of the gene products rather than by purely bioinformatic approaches.

3. The crystal structures have a high resolution, but are poorly refined. The Rfrees are 2-4% higher than where they should be at 1.6 Å. The number of Ramachandran outliers is larger than 1%, but should be smaller than 0.1%, possibly zero. The authors have to deposit more accurate structural models. The resolution enables them to do so.

We are very grateful for the effort by the reviewer to scrutinize our structural data and detect mistakes in the table presenting the statistical details. The resolutions of the crystal structures are 1.6, 1.7 and 1.85 Å for APO, ADPR-Nam, BAD, respectively. Unfortunately, when we prepared the statistics table, the table should have contained “Ramachandran allowed” instead of “Ramachandran outliers”, and “Ramachandran outliers” instead of “Rotamer outliers”. These mistakes have now been corrected and the structures of APO and ADPR-Nam have no Ramachandran outliers while BAD has only one outlier within 418 residues (0.2%). We agree, if we had a higher number of Ramachandran outliers the data would have required better refinement.

Concerning the R-factors, all structures have lower R-factors compared to the average R-factors of structures with similar resolution. We admit that for the highest resolution (1.6Å) structure one could expect lower values. However, we have not managed to refine the data any further. This can in part be explained by the fact that crystallization was performed with the construct harboring the C-terminal expression tag (13 residues including the His-tag). This tag is not modelled into the structure as the difference density ends after 233aa, however there are a few “undefined areas” of difference density close to the protein surface (between symmetry related molecules), and in close proximity to the visible C-terminus of the model. These “undefined areas” of density are not clear enough to build the amino acid chain, and none of the buffer or crystallization solution components fit the density. There is no corresponding undefined difference density visible for ADPR-NAM and BAD structures that have a slightly lower resolution. Furthermore, changing refinement program from PHENIX to Refmac provided only marginally better R-factors but no changes to the model itself.

Other comments/issues:

1. Why is there no antitoxin? Probably the presence of the NADase on the spore surface poses no risk for the viability of the spore. This is in contrast to the NADases in bacteria. This should be discussed.

We agree with the reviewer that the suspected lack of an antitoxin (referred to as immunity factor in the manuscript) is very interesting. We have now added a short discussion in the manuscript.

2. The “exchange of the nicotinamide moiety by nicotinic acid” is rather a hydrolysis of nicotinamide.

Unfortunately, we are not entirely sure what the reviewer refers to. ADP-ribosylcyclases use a catalytic mechanism that enables the exchange of the pyridine moiety. In this base-exchange reaction there is no hydrolysis of nicotinamide. For example, in the *Aplysia californica* NAD cyclase enzyme reaction the nicotinamide-ribosyl bond is cleaved via a dissociative process with a late transition state, leading to an oxocarbenium ion reaction intermediate stabilized by the side chain of an invariant glutamic acid. This step is followed by two different nucleophilic reactions in competition: (1) an intermolecular pathway involving a rapid trapping from the b-face of this intermediate by a water molecule (NAD⁺ glycohydrolases activity) or by competing neutral nucleophiles such as nicotinic acid (base exchange, which is a transglycosidation reaction). (2) An intramolecular reaction between N1 of the adenine ring and C19 (anomeric carbon) of the oxocarbenium ion leading to the formation of cyclic ADP-ribose.

3. The unit of turnover rates is per second (small “s”).

This has been corrected in the manuscript.

Reviewer #3 (Remarks to the Author)

In this study, the authors have identified NADase activity associated with the conidial surface of *Aspergillus fumigatus*, an airborne fungal pathogen, obtained it in its recombinant form by expressing the responsible gene in two different systems (insect Sf9 cells and human 293 cells) and deduced its crystal structure. They took the advantage of several biophysical techniques to study the activity as well as the structure of NADase. The structural elucidation of AfNADase is well performed and detailed. Based on the activity observed and the sequence similarity with other microbial toxins with NADase activity, the authors propose conidial NADase as the possible virulence factor mediating/facilitating fungal infection. Their speculation is based on the depletion of NAD⁺ leading to cell death, as fungal NADase lack additional cyclase activity. However, the experimental evidence proving their speculation is lacking in this study, and they do not discuss about alternative role played by ADPR/Nam, the products of fungal NADase activity.

We thank the reviewer for emphasizing the novelty and technical quality of our study. The main focus of our study was the biochemical characterization of fungal NADases. To appropriately characterize the biological role of AfNADase during interaction with the host or other microorganisms represents a study of its own. However, we have toned down the statements about the putative role of AfNADase in virulence.

Major comments:

1. The authors have used CEA17ΔakuB as the parental strain to generate ΔnadA mutant. Did they check NADase activity with other clinical isolates? Moreover, the authors have grown *A. fumigatus* on minimal medium to obtain conidia; nevertheless, is conidial associated NADase activity universally present whatever the culture medium used for conidiation? Because, fungal surface architecture as well as its secretome is variable with the growth medium/culture conditions.

We agree with the reviewer and have checked for activity on clinical isolates. This is now shown in supplemental figure 1. Moreover, we screened for activity on conidia grown on different substrates.

Indeed, as now also shown in supplemental figure 1, although there are differences in NADase activity dependent on the culture conditions NADase activity was found for all media tested.

2. What about the expression of NADA at different growth stages/morphotypes? Was it only conidia associated, or secreted/released into the culture medium during growth?

We are grateful for this important comment! The original description of *N. crassa* NADase referred to conidia only, and thus the activity was regarded as “conidial” (ref. 13). As we now show in Fig. 1B, NADase activity is present at all growth stages. The manuscript has been updated, including the title, to reflect this new information. Regarding the release of the enzyme this needs to be properly assessed, in some instances we observed activity in the medium but have not studied this systematically yet.

3. Why the NADase activity is stable even at 95 °C, is it because of the short inactivation time of 5 minutes? If *A. fumigatus* NADase functions as a dimer, then this heat treatment should destroy the quaternary structure of NADase, as the differential scanning calorimetry in Extended data figure 1 indicates the Tm of AfNADase to be around 78 °C.

The short inactivation time of 5 minutes may be the reason we still observe activity. We have performed an additional experiment showing that the enzyme is also active after 10 minutes at 95 °C, see supplemental figure 2H (previously supplemental figure 1). These results might be due to refolding or a lack of complete denaturation or a combination of both. These results show that the protein is unusually heat tolerant and may be able to refold once it has been denatured. In support of this notion, the zymograms (activity gels) shown in Fig. 1 were obtained following a renaturation procedure. Since the active protein is a homodimer, the subunits migrate in the same position and can re-assemble even following (non-reducing) SDS-PAGE.

4. 1H-NMR: the reported chemical shifts (ppm) for NAD H2, H4 and H6 are around 9.3, 8.8 and 9.1, respectively, which looks shifted specifically for H4 in the present study. How do the authors explain this discrepancy in the chemical shift?

We thank the reviewer for making us aware of this mistake! When the figure was prepared, we did not calibrate the axis and omitted setting DSS as 0 ppm. We apologies for this mistake. The corrected spectra give ppm values of 9.322, 9.135 and 8.822 for H2, H4 and H6, respectively. These values are in line with what has been reported in the human metabolome database. The same error was present in the product spectra, and the corrected values are similar to reported values.

5. Even in the presence of EGTA, there is significant NADase activity (Figure 2B), which means the activity is only partially Ca²⁺ dependent or is it because only one concentration of EGTA was tried for the assay? Moreover, even after the mutation of D219 and E220, although there was seven-fold decrease, there was still significant activity compared to control. Probably these mutations may alter the structure leading to the observed decrease in the activity than the defective Ca²⁺ binding resulting in the decreased activity.

We agree that the enzyme is only partially dependent on Ca²⁺ as there is activity both after treatment with EGTA and in the calcium binding site mutant. However, we did not mean to imply that enzyme activity is fully dependent on Ca²⁺ binding.

We have now performed an additional experiment showing that the activity is not decreased further when the concentration of EGTA is increased to 5 mM. Furthermore, the activity cannot be increased by adding more Ca^{2+} . These results demonstrate that the activity of the enzyme is only partially dependent on calcium ions. The loss of the calcium ion by chelation or mutagenesis may cause structural changes in the enzyme that allosterically regulates the enzyme activity. However major changes in the protein structure, such as unfolding or partial unfolding, does not seem to happen as the migration in the size exchange column is comparable between the calcium-binding mutant and wild-type enzyme.

6. Why the conserved residues R129, R148 and F130, if important, were not attempted to mutate to establish their importance?

We have now mutated these residues and the results are shown in Fig 4G. These new results confirm the predictions regarding the importance of these amino acids for catalysis. The R148 mutant could not be expressed in 293 cells as no protein could be detected by western blot or activity measurements in the medium, cell lysate or the insoluble fraction. R148 forms a salt bridge with D128 that may be important for protein stability. We therefore suspect that the protein is readily degraded. Interestingly, it has been reported that mutating the corresponding R780 in TNT also led to protein instability. (10.1074/jbc.RA118.005832)

7. Macrophages and neutrophils being the major innate immune cells against *A. fumigatus*, how differentially the CEA17 Δ akuB and Δ nadA mutant conidia will interact with them? What will be their virulence in vivo model?

We agree with the reviewer that this is an important question that needs thorough investigation. To appropriately address the potential role of fungal NADases in virulence will be an extensive study of its own. Therefore, we focused the present study on the discovery and comprehensive protein-chemical, kinetic and structural characterization of this protein family.

8. ADP-ribose, upon polymerization, is also known to regulate many physiological processes. ADPR-polymer is also a potent activator of proinflammatory cytokines. Therefore, how do the authors rule out the possibility of fungal NADase function in providing the substrate ADPR for host ADPR-polymerases for host-benefits?

We agree with the reviewer that ADPR-polymers regulate many physiological processes, including the activation of proinflammatory cytokines. However, as far as we know, NAD^+ , and not ADPR, is the substrate for ADPR-polymerases. Thereby, fungal NADases may prevent ADPR-polymer formation by depleting host cells of NAD^+ , which in turn might help the fungus to suppress the immune system.

Minor points:

1. Title: The authors have identified only one NADase in *Aspergillus fumigatus*, a pathogenic fungus; but the other NADase is from *Neurospora crassa*, which is more of a model fungus. Therefore, a general title, 'a new protein family in pathogenic fungi', is not appropriate.

Based on the available phylogenetic information, of course, we cannot rule out the possibility that NADases may also be present in non-pathogenic species. However, the distribution we established so far (Fig. 5D) is very suggestive. We also note that potential pathogenicity is, perhaps, not established for various species. For example, it has been reported only recently that *Neurospora crassa* is a hitherto unsuspected potential pathogen for pine trees (<https://doi.org/10.1038/srep05135>). For these reasons,

we would prefer keeping the title regarding the occurrence of the NADases in pathogenic species to alert scientists that could consider our observations in virulence studies.

2. Line 18: Expand NAD in the abstract.

The manuscript has been changed accordingly.

3. Line 19: how do the pathogens inject NADases; is 'release' a better word? Or, anything known about the mechanism of injecting NADases by pathogens?

The sentence has been changed in the manuscript.

4. Lines 21-22: the authors have found only one NADase on the conidial surface, therefore 'NADases' and 'these' may not be appropriate.

We have also shown that the *Nc*NADase enzyme is located on the surface of conidia, see Figure 1. In addition, we have shown, by sequence similarity, that the enzyme is present in a wide range of fungal species. Therefore, we believe the phrasing is appropriate.

5. Line 42: expand TIR domain.

The manuscript has been changed accordingly

6. Lines 46-48: the statements are contradictory; the authors mention no NADase has been identified in fungi, but then they mention that NADase activity has been reported in *Neurospora crassa*. Indeed, Ref. 13 is about purification and characterization of NADase (NAD⁺-glycohydrolase) from the conidia of *Neurospora crassa*.

We believe that "identified" implies the molecular identification, that is, the establishment of DNA and amino acid sequences. This has not been done for any fungal NADase so far. Indeed, the *N. crassa* enzyme has been partially purified and characterized previously, as described in reference 13. However, they did not determine the sequence, gene or protein responsible for this activity which we have done for the first time.

7. Figure 1D: *A. californica* cyclase has been described in the methods as a positive control for HPLC, but not in this figure legend.

The manuscript has been changed accordingly

8. Figure 1F: NADase activity of the complementation strain is lacking.

We have now added the complementation strain to the experiment (Fig. 1F).

9. Methods, fungal strains, media and growth conditions: replace 'formation of conidia' with 'conidiation'.

The manuscript has been changed accordingly.

10. In the methods section, the details of in gel NADase activity is not clear enough; *N. crassa* and *A. fumigatus* conidia (both heated and non-heated) were mixed with non-reducing SDS-PAGE sample buffer for how long, to see NADase activity?

The samples were mixed for 5-10 minutes with the sample buffer before being loaded on the SDS-PAGE. The section has been modified in order to avoid any confusion.

11. Substrate specificity by HPLC: is there specific reason for performing the reaction at 20 °C ? Provide a reference for visualization of the compound running through the HPLC column at 259 nm and 340 nm.

There is no specific reason why the reactions were performed at 20 °C other than using the same condition throughout. Regarding the use of different wavelengths, we agree that the detection of reduced pyridine nucleotides at both 259 nm and 340 nm may be useful in complex samples. However, here we use single substrates (either NAD⁺, NADH, NADP⁺ or NADPH). Consequently, the presentation of the 340 nm traces would not add any further information. Therefore, we did not include them in the figures.

12. Why there is discrepancies in the molecular weight of AfNADase expressed in 293 cells by size exclusion chromatography and through partition coefficient?

We agree that the estimated molecular masses do not match perfectly, but we would not really expect that either. Both SDS-PAGE and size exclusion chromatography (SEC) of glycoproteins provide only approximate molecular weight and the discrepancy is due to inherent properties of the two methods. Glycoproteins will display decreased migration in SDS-PAGE because of the poor glycan-SDS interaction. The molecular weight determined by SEC will deviate even further from the estimated value. The hydrodynamic radius rather than the mass determines the SEC elution time, additionally, the elution time is influenced by factors such as the geometric shape that cannot be corrected for by the protein standard. Furthermore, the contribution of the glycan to the hydrodynamic radius might be disproportionate leading to an exaggeration of the molecular weight due to longer retention.

13. One of the Ca²⁺ binding residue has been mentioned as S219 in the text, but in the Figure 3 as S216; which one is correct?

Thank you for pointing out this mistake! Serine 216 is correct; the text has been changed accordingly.

14. Why there are two (may be three) bands after PNGase treatment of NcNADase (Extended data figure 5)?

The NcNADase is predicted to contain three asparagine residues that are N-linked glycosylated and the three bands represent most likely successive de-glycosylation at the different sites.

15. Whether the tuberculosis necrotizing toxin domain was absent in the NcNADase?

The TNT domain is present in the NcNADase and this is indicated both in the running text and in figure 5A.

16. Is *Nc*NADase activity Ca^{2+} independent?

We tested the Ca^{2+} and EGTA sensitivity of *Nc*NADase and found that its activity is independent of the presence of these agents (supplemental Fig. 2I). These observations are consistent with the absence of the calcium binding site present in *Af*NADase.

17. What is the importance of *Nc*NADase, as it is a saprophyte.

As commented above, while *Neurospora crassa* is a saprophyte it has also been reported to be pathogenic towards Scots pine (*Pinus sylvestris*), perhaps *Nc*NADase plays a role in this pathogenic process.

18. *Af* and *Nc* should be in italics before NADase, as they represent fungal species.

The manuscript has been changed accordingly.

19. Is there any detail about the glycosylation pattern of the recombinant proteins? Are they not important for NADase activity?

Mass spectrometry and the crystal structure revealed that *Af*NADase is N-linked glycosylated at three asparagine residues. The length and branching of the polymers of native conidial *Af*NADase seems to be more extensive compared to the recombinant protein expressed in both 293 and Sf9 cells based on their SDS-PAGE migration. We do not know whether glycosylation is important for activity. However, it may be important for protein stability as the protein expresses as inclusion bodies in *E. coli* (data not shown).

20. Whether the NADase activity of CpnT is Ca^{2+} dependent?

There is no reported data suggesting the activity of CpnT to be Ca^{2+} dependent. Based on the alignment shown in supplemental Figure 8, the protein lacks the C-terminal calcium ion binding site found in *Af*NADase. As we show, the *Nc*NADase also lacks the Ca^{2+} binding site and is insensitive to calcium ions. Therefore, it is very likely that CpnT is indeed Ca^{2+} insensitive.

Reviewer comments, second round:

Reviewer #1 (Remarks to the Author):

The authors fully addressed my comments.

Reviewer #2 (Remarks to the Author):

The authors addressed all critiques adequately, with one exception. Clearly, the authors identified a fungal NADase. This is the main function of the protein and there is significant and identifiable sequence similarity of this core domain to bacterial TNT proteins. Furthermore, the structures of these core domains are identical as are the active sites of TNT and of the NADase of *A. fumigatus*. Clearly, these NADases belong to the same class as also recognized by the Pfam algorithms. This issue needs to be recognized throughout the manuscript. Additional domains are also present in bacterial NADases of the TNT family depending on their subcellular localization or secretion mechanisms, but the core domain defining the function and the protein family is the same. The emphasis of the manuscript needs to be on the structure of the enzyme in complex with a non-hydrolyzable NAD analog which leads to a plausible mechanism of NAD hydrolysis for the TNT family of NADases and the novel features of the subfamily of NADases of fungal spores. This issue needs to be fixed by editing the summary, introduction and interpretation. Consequently, the extensive description of the experimental discovery of the AfNADase (pages 2-3) needs to be shortened drastically and the figures 1D, 1E should be moved to the supplement.

Reviewer #3 (Remarks to the Author):

With the modifications made and incorporating the additional information both descriptive-wise as well as experimentally, the manuscript reads well. I am convinced with the answers provided by the authors to the queries raised from my part. Importantly, the revision made about AfNADase and virulence is appropriate as the study is now about structural elucidation of the AfNADase.